# Risk Sensitive Dead-end Identification in Safety-Critical Offline Reinforcement Learning

**Taylor W. Killian**                                        *twkillian@cs.toronto.edu*
*University of Toronto, Vector Institute*
*Massachusetts Institute of Technology*

**Sonali Parbhoo**                                             *sparbhoo@imperial.ac.uk*
*Imperial College London*

**Marzyeh Ghassemi**                                            *mghassem@mit.edu*
*Massachusetts Institute of Technology*
*CIFAR AI Chair, Vector Institute*

**Reviewed on OpenReview:** *https://openreview.net/forum?id=oKlEOT83gI*

## Abstract

In safety-critical decision-making scenarios being able to identify worst-case outcomes, or dead-ends is crucial in order to develop safe and reliable policies in practice. These situations are typically rife with uncertainty due to unknown or stochastic characteristics of the environment as well as limited offline training data. As a result, the value of a decision at any time point should be based on the *distribution* of its anticipated effects. We propose a framework to identify worst-case decision points, by explicitly estimating *distributions* of the expected return of a decision. These estimates enable earlier indication of dead-ends in a manner that is tunable based on the risk tolerance of the designed task. We demonstrate the utility of Distributional Dead-end Discovery (DistDeD) in a toy domain as well as when assessing the risk of severely ill patients in the intensive care unit reaching a point where death is unavoidable. We find that DistDeD significantly improves over prior discovery approaches, providing indications of the risk 10 hours earlier on average as well as increasing detection by 20%.

## 1  Introduction

In complex safety-critical decision-making scenarios, being able to identify signs of rapid deterioration is crucial in order to proactively adjust a course of action, or policy. Consider the challenge of replacing an aging component within high-value manufacturing machinery. The longer one waits to replace this component, the efficiency of the process degrades until catastrophic failure at some unknown future time. However, the cost of temporarily stopping manufacturing to replace the component is non-trivial and the observed state of the system may not transparently signal when failure is imminent. Specifically, being aware of potential "worst-case" outcomes when choosing whether to delay repair is paramount to develop both safe and successful policies. Yet quantifying the worst-case outcomes in these and related circumstances among other safety critical domains–such as healthcare–is usually challenging as a result of unknown stochasticity in the environment, potentially changing dynamics, limited data, and the wide range of possible outcomes that might follow a sequence of decisions. By reliably providing an early indication of system failure to human operators, they would be enabled to intervene and make the necessary repairs in order to avoid system failure.

Reinforcement learning (RL) is a natural paradigm to address sequential decision-making tasks in safety-critical settings, focusing on maximizing the cumulative effects of decisions over time (Sutton & Barto, 2018).

RL frameworks have been posed to design safe and responsible machine learning algorithms by regulating undesirable behavior with safety tests (Thomas et al., 2019) or through establishing performance guarantees when learning from limited data (Liu et al., 2020). Unfortunately, these approaches to develop safe RL policies depend on the ability to characterize *a priori* what actions or regions of the state space to avoid. This is not feasible in many real-world tasks as the definition of unsafe or risky behaviors may not be tractable due to unknown interactions between the observed state and selected actions.

A defining feature of RL in high-risk real-world settings is that the learning paradigm is fully *offline* **and** *off-policy* since exploratory data collection is often infeasible due to legal, safety, and ethical implications. However, RL methods are heavily influenced by the data collection policy: data is collected prior to learning, and frequently contains decisions that rely on confounding information; such as production schedules requiring deviations from normal use of manufacturing machinery or lifestyle information and insurance status in clinical treatment (Dorfman et al., 2021; Gasse et al., 2021). These factors, if unaccounted for, may lead to the overestimation of the anticipated return, biased decisions, and/or overconfident yet erroneous predictions (Thrun & Schwartz, 1993). In addition, rare but dangerous situations can be overlooked if optimizing without accounting for possible "worst case" outcomes, thus failing to guarantee safety.

While RL has been explored within healthcare applications, it has primarily been used as a means for learning *risk-neutral* policies, optimized to provide the action with highest expected return (Raghu et al., 2017; Parbhoo et al., 2017; Prasad et al., 2017; Yu et al., 2021). Without the ability to explore or otherwise test alternative treatment strategies, the learned policies are unreliable (Gottesman et al., 2019; Oberst & Sontag, 2019). An alternative offline RL paradigm was introduced by Fatemi et al. (2021) that prioritizes the avoidance of actions, proportional to their risk of leading to dead-ends (where an agent enters an irrecoverably negative trajectory). In their proposed dead-end discovery (DeD) framework, recorded negative outcomes are leveraged to identify behaviors that should be avoided. Specifically, actions that lead to dead-ends are identified based on thresholded point-estimates of the expected return of that action rather than considering the full distribution. In doing so, risk estimation in DeD is limited and, at worst, too optimistic to determine which actions are safe to be executed. The implications of this are significant: by underestimating the risk associated with a particular action, we are unable to determine whether an action could be potentially dangerous – a necessity in safety-critical settings.

In this paper, we propose a risk-sensitive decision-making framework positioned to serve as an early-warning system for dead-end discovery. Broadly, our framework may be thought of as a tool for thinking about risk-sensitivity in data-limited offline settings. Our contributions are as follows: (i) Unlike former approaches, we incorporate distributional estimates of the return (Bellemare et al., 2022) to determine when an observed state is at risk of becoming a dead-end from the expected worst-case outcomes over available decisions (Chow et al., 2015). (ii) We establish that our risk-estimation procedure serves as a lower-bound to the theoretical results underlying DeD (Fatemi et al., 2021), maintaining important characteristics for assessing when identifying dead-ends. As a result, we are able to detect and provide earlier indication of high-risk scenarios. (iii) By modeling the full distribution of the expected return, we construct a spectrum of risk-sensitivity when assessing dead-ends. We show that this flexibility allows for tunable risk estimation procedures and can be customised according to the task at hand. (iv) Finally, we provide empirical evidence that our proposed framework enables an earlier determination of high-risk areas of the state space on both a simulated environment and a real application within healthcare of treating patients with sepsis.

## 2 Related Work

**Safe and Risk-sensitive RL** A shortcoming of most approaches to offline RL is that they are designed to maximise the expected value of the cumulative reward of a policy. This assumes that the training data is sufficient to promote convergence toward an optimal policy. As a result they are unable to quantify the risk associated with a learnt policy to ensure that it acts in the intended way. The field of safe RL instead tries to learn policies that obtain good performance in terms of expected returns while satisfying some safety constraints during learning and/or deployment (García & Fernández, 2015), defined through a constrained MDP (CMDP). Several safe RL algorithms (Achiam et al., 2017; Berkenkamp et al., 2017; Alshiekh et al., 2018; Tessler et al., 2019; Xu et al., 2021; Yang et al., 2022; Polosky et al., 2022) have been developed

that either i) transform the standard RL objective to include some form of risk or, ii) leverage external knowledge to satisfy certain safety constraints and quantify performance with a risk metric. However, safe RL assumes *a priori* knowledge of what unsafe regions are–through the definition of constraints whether implicitly through the environment or explicitly through agent behavior design–which is not always feasible in real-world safety-critical scenarios. Unlike these, we do not explicitly learn a policy, but learn a value function that conveys the risks inherent in making suboptimal decisions at inopportune times.

Risk-sensitive RL instead focuses on learning to act in a dynamic environment, while accounting for risks that may arise during the learning process (Mihatsch & Neuneier, 2002), where high risk regions do not have to be known a priori. Unlike risk-neutral RL, these methods optimise a *risk measure of the returns* rather than the average or expected return. Among these, Fu et al. (2018) present a survey of policy optimization methods that consider stochastic formulations of the value function to ensure that certain risk constraints may be satisfied when maximising the expected return. Other approaches propose replacing the expected long-term reward used by most RL methods, with a *risk-measure* of the total reward such as the Conditional-Value-at-Risk (CVaR) (Chow et al., 2015; Stanko & Macek, 2019; Ying et al., 2022; Du et al., 2022) and develop a novel optimization strategy to minimize this risk to ensure safety all-the-time. Ma et al. (2021) adapt distributional RL frameworks (Bellemare et al., 2022) to offline settings and by penalizing the predicted quantiles of the return for out-of-distribution actions. While these methods may be used to learn a distribution of possible outcomes, they have not been used to identify dead-ends as we propose here.

Unlike off-policy evaluation methods, we focus on estimating the *risk* associated with a policy in terms of the expected worst case outcomes. Specifically, we learn a distributional estimate of the future return of a policy using Implicit Quantile Networks (IQN) (Dabney et al., 2018), and integrate a conservative Q-learning (CQL) penalty (Kumar et al., 2020) into the loss to lower bound on the expected value of the policy.

**Nonstationary and Uncertainty-Aware RL**   Several works focus on explicitly modelling *non-stationary dynamics* in MDPs for decision-making that accounts for uncertainty over model dynamics. Among these, methods such as Chandak et al. (2020) focus on safe policy optimization and improvement in non-stationary MDP settings. Here, the authors assume that the non-stationarity in an MDP is governed by an exogenous process, or that past actions do not impact the underlying non-stationarity. Sonabend et al. (2020) use hypothesis testing to assess whether, at each state, a policy from a human expert would improve value estimates over a target policy during training to improve the target policy. More recently, Joshi et al. (2021) presented an approach for learning to defer to human expertise in nonstationary sequential settings based on the likelihood of improving the expected returns on a particular policy. Our work differs from these in that instead of focusing on optimizing a specific policy, we explicitly learn which types of behaviors to avoid using risk-sensitive distributional estimates of the *future return*, as opposed to a point estimate of the expectation of that distribution.

**RL in safety critical domains**   There are several works posed for uncertainty decomposition in applications such as healthcare. Specifically, Depeweg et al. (2018), decompose the uncertainty in bayesian neural networks to obtain an estimate of the aleatoric uncertainty for safety. Similarly, Kahn et al. (2017) use uncertainty-aware RL to guide robots to avoid collisions, while Cao et al. (2021) develop a domain-specific framework called Confidence-Aware RL for self-driving cars to learn when to switch between an RL policy and a baseline policy based on the uncertainty of the RL policy. Unlike these works, we propose a general purpose framework that can be applied to a number of safety-critical applications using risk-sensitive RL to provide an early warning of risk over possible future outcomes.

## 3   Preliminaries

As outlined above, we frame risk identification for safety critical decision making within a Reinforcement Learning (RL) context. We consider a standard episodic RL setting in an environment with non-stationary and stochastic dynamics where an agent determines actions $a \in \mathcal{A}$ after receiving a state representation $s \in \mathcal{S}$ of the environment, modeled as a Markov Decision Process (MDP) $\mathcal{M} = \{\mathcal{S}, \mathcal{A}, T, R, \gamma\}$, where $T(\cdot|s, a)$ relates to the stochastic transition from state $s$ given action $a$; $R(s, a)$ is a finite, binary reward function that provides reward only at the terminal state of each episode and $\gamma \in (0, 1]$ is a scalar discount factor. In offline

safety critical settings, we assume that recorded actions are selected according to an unknown expert policy $\pi(\cdot|s)$, given the observed state $s$. The objective is to estimate the value of each action as the discounted sum of future rewards (e.g. the return) $Z^\pi(s,a) = \sum_{t=0}^\infty \gamma^t R(s_t, a_t)$ where $s_0 = s$, $a_0 = a$, $s_t \sim T(\cdot|s_{t-1}, a_{t-1})$, and $a_t \sim \pi(\cdot|s_t)$. By characterizing the full probabilistic nature of how $Z^\pi(s,a)$ can be computed, it is used to represent the distribution of future return from the state $s$, executing action $a$.

**Distributional RL** In challenging real-world scenarios, the consequences of a decision carry a measure of unpredictability. Standard approaches to RL seek to maximize the mean of this random return. In reality, complex phenomena in stochastic environment may fail to be accounted for, leading to rare but critical outcomes going ignored. To account for this, Distributional RL (Bellemare et al., 2022) has been introduced to model the full return distribution by treating the observed return from following a policy $\pi$ and associated states as random variables when forming the Bellman equation:

$$Z^\pi(s,a) \stackrel{D}{=} R(s,a) + \gamma Z^\pi(s', a')$$

The return distribution $Z^\pi(s,a)$ is most commonly represented in RL by the state-action value function $Q^\pi(s,a)$ which represents the expected future return. That is, $Q^\pi(s,a) = \mathbb{E}[Z^\pi(s,a)]$.

As the distribution is an infinite dimensional object, some approximations are needed for tractable estimation. Initially, the support of the distribution was discretized *a priori* over pre-defined categorical quantiles (Bellemare et al., 2017). More recently this approximation has been relaxed to a distribution of uniformly weighted particles, estimated with neural networks (Dabney et al., 2018), to implicitly represent these quantiles.

Given the flexibility of these implicit quantile networks (IQN), they are well suited to define risk-aware decision criteria over value functions learned from real-world data where the anticipated return structure is unknown. As such, we build our proposed framework from IQN estimates of the state-action value function.

**Conservatism in offline RL** An important consideration when learning from offline data with RL is avoiding value overestimation for actions not present in the data (Fujimoto et al., 2018; 2019; Bai et al., 2022). Prior work has attempted to choose a lower bound of approximated value functions (Fujimoto et al., 2018; Buckman et al., 2020), regularize policy learning by the observed behavior (Fujimoto et al., 2019; Wu et al., 2019; Kumar et al., 2019; Wang et al., 2020) or by directly regularizing the value estimates of the observed actions (Kumar et al., 2020; Jin et al., 2021). We utilize this last approach (termed conservative Q-learning; CQL) which resorts to minimizing the estimated values over the observed state-action distribution to enforce a conservative lower-bound of the value function. This is accomplished by simply adding a $\beta$-weighted penalty term $\mathcal{L}^{\mathrm{CQL}}$ to the RL objective $\mathcal{L}^{\mathrm{RL}}$. Thereby the optimzation objective becomes

$$\mathcal{L}^{\mathrm{RL}} + \beta \mathcal{L}^{\mathrm{CQL}}$$

where $\mathcal{L}^{CQL}$ is chosen to be exponentially weighted average of Q-values for the chosen action (CQL($\mathcal{H}$) in Kumar et al. (2020)). This serves to additionally constrain the overestimation of actions not present in the dataset and has been shown to improve risk-averse performance with distributional RL (Ma et al., 2021). By increasing the value of $\beta$, the overall conservatism and thus risk-aversion is increased as the optimization of the estimated values is constrained further from the true value function.

### Risk estimation.

We assume the return is bounded (e.g. $\mathbb{E}[|Z|] < \infty$) with cumulative distribution function $\mathcal{F}(z) = \mathbb{P}(Z \leq z)$. When estimating the possible effects of a decision, we want to account for worst-case outcomes that occur with some level of confidence $\alpha \in (0,1)$. The value-at-risk (VaR) with confidence $\alpha$ represents the $\alpha$-quantile of the distribution $Z$: $\mathrm{VaR}_\alpha(Z) = \min\{z \mid \alpha \leq \mathcal{F}(z)\}$. This quantile can then be used to determine the "expected worst-case outcome", or conditional value at risk (CVaR):

$$\mathrm{CVaR}_\alpha(Z) = \frac{1}{\alpha}\mathbb{E}[(Z - \mathrm{VaR}_\alpha(Z))^-] + \mathrm{VaR}_\alpha(Z)$$

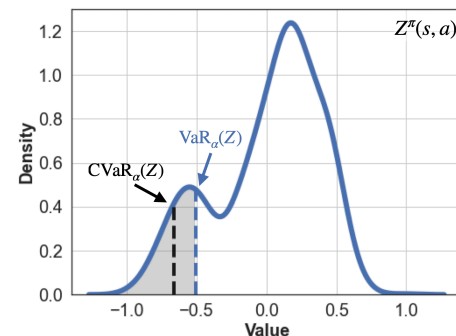

Figure 1: Illustration of the determination of conditional value at risk ($\mathrm{CVaR}_\alpha$), with $\alpha = 0.1$

where $(x)^- = \min(x, 0)$ is the negative part of $x$. We use
the dual representation of CVaR (Artzner et al., 1999) which is formulated with a single expectation:

$$\text{CVaR}_\alpha(Z) = \min_{\xi \in \mathcal{U}_{\text{CVaR}}(\alpha, \mathbb{P})} \mathbb{E}_\xi[Z]$$

where $\mathbb{E}_\xi[Z]$ is the $\xi$-weighted expectation of $Z$ within the $\alpha$-quantile and $\mathcal{U}_{\text{CVaR}}(\alpha, \mathbb{P})$ is the portion of $Z$ that falls below $\text{VaR}_\alpha(Z)$. This establishes that:

$$\text{CVaR}_\alpha(Z) \leq \mathbb{E}[Z] \tag{1}$$

as $\alpha \to 1$, then $\mathcal{U}_{\text{CVaR}}(\alpha, \mathbb{P})$ encompasses all of $Z$ and $\text{CVaR}_\alpha(Z) \to \mathbb{E}[Z]$. Thus, the CVaR is a lower-bound for value estimates derived through the expectation of the return distribution (e.g. the value function $Q^\pi$).

**Dead-end Discovery (DeD).** As introduced by Fatemi et al. (2021) the DeD framework assures a notion of security when estimating whether an action will lead to a dead-end (see Eqt. 2). DeD constrains the scope of a given policy $\pi$ if *any* knowledge exists about undesired outcomes. Formally, if at state $s$, action $a$ transitions to a dead-end at the next state with probability $P_D(s, a)$ or the negative terminal state with probability $F_D(s, a)$ with a level of certainty $\lambda \in [0, 1]$, then $\pi$ must avoid $a$ at $s$ with the same certainty:

$$P_D(s, a) + F_D(s, a) \geq \lambda \implies \pi(s, a) \leq 1 - \lambda. \tag{2}$$

Note that a dead-end may occur an indeterminate number of steps prior to the negative terminal condition. The defined notion of a dead-end is that once one is reached, all subsequent states are also dead-ends up to and including the negative terminal state. While $P_D$, $F_D$, and $\lambda$ may not be able to be explicitly calculated, the DeD framework learns an estimate of the likelihood of transitioning to a dead-end as well as the reduction in likelihood of a positive outcome. This is done by constructing two independent MDPs $\mathcal{M}_D$ and $\mathcal{M}_R$ from the base environment MDP $\mathcal{M}$ focusing solely on negative and positive outcomes, respectively. DeD learns value approximations of each MDP, $Q_D(s, a)$ for negative outcomes and $Q_R(s, a)$ for positive outcomes ($Q_D \in [-1, 0]$ and $Q_R \in [0, 1]$ respectively). These value estimates enable the identification and confirmation of dead-ends and actions that lead to them through the relationship:

$$-Q_D(s, a) \geq P_D(s, a) + F_D(s, a) \tag{3}$$

Then, the security condition is assured by $\pi(s, a) \leq 1 + Q_D(s, a)$. In practice, the $Q_D$ and $Q_R$ functions are approximated with deep Q-networks (DQN) (called the D- and R- networks, respectively) in concert with empirically determined thresholds $\delta_D$ and $\delta_R$ to flag when actions or states have the risk of leading to dead-ends and should be avoided.

The DeD framework determines an action $a$ should be avoided when both $Q_D(s, a) \leq \delta_D$ **and** $Q_R(s, a) \leq \delta_R$. A state $s$ is said to be a dead-end if the *median* value over all actions falls below these thresholds. That is a dead-end is reached whenever both $\text{median}(Q_D(s, \cdot)) \leq \delta_D$ **and** $\text{median}(Q_R(s, \cdot)) \leq \delta_R$. Our proposed distributional formulation of dead-end discovery uses these definitions, with slight adaptation to the risk-sensitive approach we use, allowing for the identification of **both** high-risk actions and states. However, in this paper we prioritize the identification of dead-end states, demonstrating that our proposed solution provides earlier identification.

## 4 Risk-sensitive Dead-end Discovery

While the DeD framework is promising for learning in offline safety-critical domains, it has limited risk-sensitivity by neglecting to model the full distribution of possible outcomes. We develop a risk-sensitive framework for dead-end discovery that conservatively models the full distribution of possible returns, driven by irreducible environment stochasticity. Our approach, DistDeD, utilizes distributional dynamic programming (Bellemare et al., 2022) to estimate the full distribution of possible returns while also limiting overestimation due to out-of-distribution actions by incorporating a CQL penalty (Kumar et al., 2020).

Mirroring the construction of DeD, we instantiate two Markov Decision Processes (MDPs) $\mathcal{M}_D$ and $\mathcal{M}_R$, derived from the original MDP $\mathcal{M}$, $\gamma = 1$, with reward functions chosen to focus on either the positive or

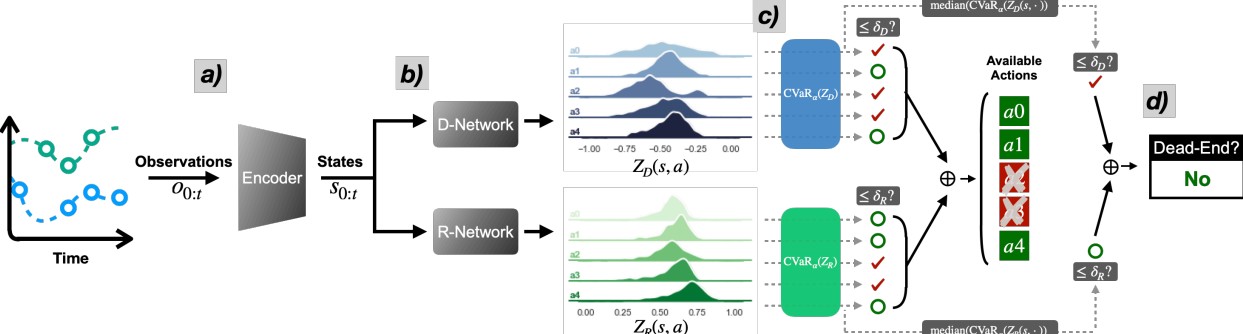

Figure 2: **Distributional Dead-end Discovery (DistDeD)** a) Observations are encoded (as needed) into a state representation and then b) passed to independent IQN models to estimate the distribution of returns ($Z_D$ and $Z_R$) for each possible action. c) The $\text{CVaR}_\alpha$ is computed for each distribution and is then evaluated against the thresholds $\delta_D$ and $\delta_R$. If both $\text{CVaR}_\alpha(Z_D)$ and $\text{CVaR}_\alpha(Z_R)$ fall below the respective thresholds for any action, then that action is recommended to be avoided. d) If the median over all actions falls below the thresholds for both distributions, then the state is said to be a dead-end.

negative outcomes. $\mathcal{R}_D$ returns $-1$ with any transition to a negative terminal state and is zero otherwise. $\mathcal{R}_R$ returns $+1$ with any transition to a positive terminal state and is zero otherwise. We then approximate the distributional returns $Z_D$ and $Z_R$ of these separate MDPs independently, where the support of $Z_D$ is $[-1, 0]$ and the support of $Z_R$ is $[0, 1]$.

To quantify the risk of selecting an action $a$ at state $s$, we consider the expected worst-case outcome—or conditional value at risk (CVaR)—of these return distributions. That is, we infer $\text{CVaR}_\alpha(Z_D(s, a))$ and $\text{CVaR}_\alpha(Z_R(s, a))$ for a chosen $\alpha \in (0, 1]$, which we consider to be a hyperparameter along with the choice of thresholds $\delta_D$ and $\delta_R$. By using CVaR to determine the risk of approaching a dead-end, we effectively construct a lower-bound on the DeD value estimates (by virtue of Eqt. 1) which allows us to maintain the same theoretical framing. Since DeD is built around the expectation of the return: $Q_D(s, a) = \mathbb{E}[Z_D(s, a)]$. Then, as $\text{CVaR}_\alpha(Z_D(s, a)) \leq \mathbb{E}[Z_D(s, a)]$ we are assured that:

$$-\text{CVaR}_\alpha(Z_D(s, a)) \geq -Q_D(s, a) \geq P_D(s, a) + F_D(s, a) \tag{4}$$

Thus, by bounding the estimates of entering a dead-end, we see that using CVaR satisfies the security condition: $\pi(s, a) \leq 1 + \text{CVaR}_\alpha(Z_D(s, a))$. Parallel results for $Z_R$ follow similarly.

We choose to represent the distributions $Z_D$ and $Z_R$ for all states $s$ and actions $a$ using implicit Q-networks (IQN) (Dabney et al., 2018). To constrain the distributional estimates from overestimating the return for actions not present in the dataset, thus avoiding overconfidence, we train the IQN architectures with a conservative Q-learning (CQL) penalty (Kumar et al., 2020). CQL regularizes the distributional Bellman objective by minimizing the value of each action, which serves also to constrain overestimation of actions not present in the observed data. We weight this penalty by the hyperparameter $\beta$.

An illustration of the DistDeD framework is included in Figure 2: *a)* If necessary[1], observations are encoded into a state representation. *b)* The encoded state representations are then passed to independent IQN models to estimate $Z_D(s, \cdot)$ and $Z_R(s, \cdot)$ for each possible action. *c)* The CVaR is computed for each distribution and then evaluated against the thresholds $\delta_D$ and $\delta_R$. Following the definition of dead-end discovery given in the previous section, if both $\text{CVaR}(Z_D)$ and $\text{CVaR}(Z_R)$ fall below the respective thresholds for any action, that action is recommended to be avoided. *d)* Furthermore, if the median over all actions falls below the thresholds for both distributions, then the state is said to be a dead-end.

With the bounding provided by DistDeD, utilizing CVaR estimates of the inferred return distributions, we enable a more conservative and thereby risk-averse mechanism to determine whether a state $s$ is at risk of being a dead-end. The level of risk-aversion, or conservatism, is jointly determined by the confidence level

---

[1]When observations are irregular or partial

$\alpha$, the weight of the CQL penalty $\beta$ as well as the thresholds $\delta_D$ and $\delta_R$. The level of conservatism within DistDeD depends on choices of all of these quantities. Since $\beta$ directly affects the optimization process of the D- and R- Networks, we treat it as a hyperparameter. An investigation of the affect of increasing $\beta$ can be found in Section A.4.3 in the Appendix. The choice of $\alpha$, influencing the CVaR calculation, as well as the thresholds $\delta_D$ and $\delta_R$ can be tuned dependant on acceptable risk tolerances in the task when evaluating the trained D- and R- Networks. Choosing a smaller value for $\alpha$ constrains the CVaR evaluation of the estimated distributions to consider lower likelihood (and more adverse, by construction) outcomes, a form of increased conservatism. Smaller values of the thresholds increase the sensitivity of the risk determination of the framework. We demonstrate the effects of choosing different $\alpha$ values on the performance benefits of DistDeD in comparison to previous dead-end discovery approaches (Fatemi et al., 2021) across multiple settings of $\delta_D$ and $\delta_R$ in our experiments using real-world medical data in Section 6.

## 5 Illustrative Demonstration of DistDeD

We provide a preliminary empirical demonstration of the advantages seen by using our proposed DistDeD framework using the LifeGate toy domain (Fatemi et al., 2021). Here, the agent is to navigate around a barrier to a goal region while learning to avoid a dead-end zone which pushes the agent to the negative terminal edge after a random number of steps (See Figure 3).

**Empirical Comparison** We aim to demonstrate the apparent advantages of our proposed DistDeD in comparison to the original DeD framework. For DeD, we model the $Q_D$ and $Q_R$ functions using the DDQN architecture (Hasselt et al., 2016) using two layers of 32-nodes with ReLU activations and a learning rate of $1e^{-3}$. For DistDeD we utilize IQN architectures (Dabney et al., 2018) for both $Z_D$ and $Z_R$ using two layers of 32 nodes, ReLU activations and the same learning rate of $1e^{-3}$. For each IQN model, we sample $N, N' = 8$ particles from the local and target $\tau$ distributions while training and also weight the CQL penalty $\beta = 0.1$. When evaluating $Z_D$ and $Z_R$, we select $K = 1000$ particles and set our confidence level to $\alpha = 0.1$.

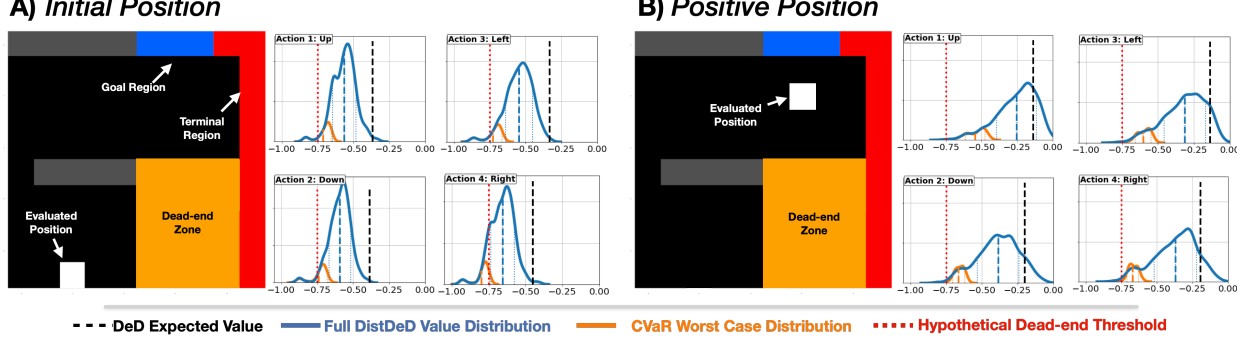

Figure 3: Demonstration of inherent value of using $Z_D(s, a)$ estimated with IQN and $\mathrm{CVaR}_{0.1}(Z_D(s, a))$ in comparison to $Q_D(s, a)$ estimated with DDQN on the LifeGate toy domain (Fatemi et al., 2021). **A)** Evaluating returns from an initial state, **B)** evaluating returns from a more favorable location near the goal region. Notably, the CVaR estimate (the mean of the orange "worst-case distribution") is *risk-sensitive* and *provides a lower bound of the expected value of the blue return distribution*, while the value estimate of DeD (black dashed line) is far more optimistic. Here, we set $\delta_D = -0.75$ as a notional threshold (red dashed line).

All approximate value functions (both expectational and distributional) were trained using 1 million randomly collected transitions from LifeGate. In Figure 3 we show the learned value estimates from the D-Networks for all actions available to the agent in select locations. We suppress the corresponding R-Network estimates for visual simplicity although they reflect qualitatively the same thing. For this demonstration we plot the full return distribution $Z_D(s, a)$, the $\alpha$-quantile used to compute $\mathrm{CVaR}_\alpha(Z_D(s, a))$, the value estimate $Q_D(s, a)$ from the DeD, as well as a notional threshold $\delta_D = -0.75$.

We see the inherent value of the distributional estimates used in DistDeD to determine which actions to avoid. Fig. 3(A) presents the returns at an initial state, from which encountering a dead-end is more common.

Fig. 3(B) presents the estimated returns from a more favorable location near the goal region. As expected, the CVaR estimate, the mean of the orange "worst-case distribution", is a lower bound on the expected value of the full return distribution (plotted in blue). Notably, the value estimated using DeD (black dashed vertical line) is far more optimistic, since DeD only considers thresholded point-estimates of expected value. This provides evidence of the limitations of DeD, ignoring the full return distribution when estimating the value of available decisions.

In Figure 4(A, B), we evaluate three pre-determined policies in LifeGate using both DeD and DistDeD. Two of the three policies attempt to navigate through the dead-end region of the environment. This construction is purposeful in order to indicate how reliably risk is flagged by each approach. The design of this experiment is to demonstrate the early-warning capability of DistDeD for those sub-optimal trajectories. In Figure 4(C) we evaluated 10,000 trajectories with stochastic execution of the two suboptimal policies and assess how many steps prior to entering the dead-end region that DistDeD and DeD raise alarm and recommend a change in policy. We assess the overall risk of each state $s$ in a trajectory by averaging the median values of $Q_D(s, \cdot)$ and $Q_R(s, \cdot) - 1$ (for DistDeD CVaR$_\alpha(Z_D(s, \cdot))$ and CVaR$_\alpha(Z_R(s, \cdot)) - 1$). If the averaged median value falls below the threshold $\delta_D$, an alarm is raised. We use the previously published value, $\delta_D = -0.15$ for DeD and choose $\delta_D = -0.5$ for DistDeD. These values were chosen empirically by attempting to minimize false-positives among a validation set of the data (see Section A.4.1 for more detail).

DeD (Fig. 4(A)) fails to adequately signal the risk of the two sub-optimal policies before they reach the dead-end region of the environment. In contrast, DistDeD (Fig. 4(B)) appropriately flags the trajectories ahead of the dead-end region, allowing for correction if an overseeing agent is able to intervene. Fig. 4(C) quantifies this advantage, demonstrating that DistDeD provides an indication of risk, on average, 3 steps earlier. This result confirms the utility of modeling the full distribution of expected returns and using a more coherent estimation of risk, focused on expected worst case outcome.

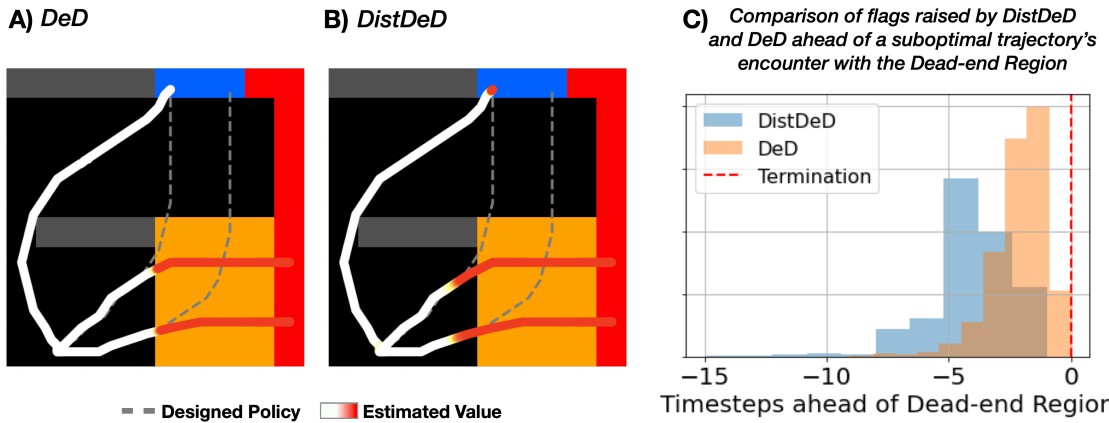

Figure 4: DistDeD's advantage when alerting that a trajectory is at risk of encountering a dead-end in the LifeGate domain. Three hand-designed policies (with two purposefully suboptimal) (shown in white) are evaluated using both DeD (A) and DistDeD (B), showing that DistDeD raises alarm earlier than DeD and in a manner that could alert a necessary change in policy before encountering a dead-end. 10000 stochastic executions of these suboptimal policies are then evaluated (C) using both approaches to understand the scope of how much earlier DistDeD raises a flag in comparison to DeD. Dotted lines show how raising alarms earlier leads to actions that could direct a patient's trajectory towards potential recovery (shown in blue).

## 6 Assessing Medical Dead-ends with DistDeD

**Data** We aim to identify medical dead-ends among a cohort of septic patients derived from the MIMIC-IV (Medical Information Mart for Intensive Care, v2.0) database (Johnson et al., 2020). This cohort comprises the recorded observations of 6,188 patients (5,352 survivors and 836 nonsurvivors), with 42 features, and 25 treatment choices (5 discrete levels for each of IV fluid and vasopressor), over time periods ranging between

12 and 72 hours. We aggregate each feature into hourly bins and fill missing values with zeros, keeping track of which features were actually observed with an appended binary mask. Missing features are implicitly accounted for when constructing state representations of a patient's health through time. Details about the exclusion and inclusion criteria used to define the construction of this patient cohort are contained in Section A.1 in the Appendix.

**State Construction** As recommended by Killian et al. (2020) and implemented in DeD (Fatemi et al., 2021), we make use of a sequential autoencoder to construct fixed dimension state representations, embedding a history of recorded observation of a patient's health previous to each time step. This allows us to process partial and irregularly occurring observations through time, a characteristic of medical data. To do this, we use an online Neural Controlled Differential Equation (NCDE) (Morrill et al., 2021) for state construction as it naturally handles irregular temporal data. Additional information about the NCDE state construction can be found in Section A.2.1 in the Appendix. We define terminal conditions for each trajectory as whether the patient survives or succumbs to (within 48 hours of the final observation) their infection. There are no intermediate rewards aside from these terminal states. When a patient survives, the trajectory is given a $+1$ reward, where negative outcomes receive $-1$.

**D- and R- Networks** The encoded state representations provided by the NCDE are provided as input to the D- and R -Networks to estimate the value (and risk of encountering a dead-end) of each state and all possible treatments. To form the DistDeD framework we use CQL (Kumar et al., 2020) constrained implementations of IQN (Dabney et al., 2018) to train each network, as discussed in Section 4 (details included in Appendix A.2.2).

**Training** We train the NCDE for state construction as well as the IQN instantiations for the D-, and R-Networks in an offline manner. All models are trained with 75% of the data (4,014 surviving patients, 627 patients who died),validated with 5% (268 survivors, 42 nonsurvivors), and we report all results on the remaining held out 20% (1,070 survivors, 167 nonsurvivors). In order to account for the data imbalance between positive and negative outcomes, we follow a similar training procedure as DeD (Fatemi et al., 2021) where every sampled minibatch is ensured to contain a proportion of terminal transitions from non-surviving patient trajectories. This amplifies the training for conditions that lead to negative outcomes, ensuring that the D- and R- Networks are able to recognize scenarios that carry risk of encountering dead-ends. Specific details on the training of DistDeD can be found in Appendix A.2.[2]

### 6.1 Experimental Setup

By design, DistDeD is formulated to provide a more conservative and thereby earlier indication of risk. A secondary benefit of the design of DistDeD is that by adapting the risk tolerance level of the CVaR estimates (by selecting different values for $\alpha$), we are provided a spectrum of value functions that could be used to assess whether a dead-end has been reached or is eminent. We therefore aim to execute a set of experiments that assess the extent at which these two points of improvement over DeD provide benefit. By establishing more conservative estimators with the IQN D- and R- Networks, we increase the occurrence of what could be identified as false positive indications of risk for patients whose health has not deteriorated to be a legitimate dead-end (e.g. patients who survive). We therefore need to assess the tradeoffs of increased "false-positives" against improved recall for indications of risk for patients who died.

To perform this assessment we execute a set of experiments to quantitatively compare DistDeD to DeD when each approach is applied to the septic patient cohort outlined above. First, this entails measuring how much earlier DistDeD raises flags across a range of VaR $\alpha$ values (for a fixed set of thresholds $\delta_{D,R}$). Second, we want to identify if DistDeD's variation—due to the choice of VaR $\alpha$—introduces settings that perform worse than DeD when considering a full range of possible thresholds $\delta_{D,R}$. Finally, we aim to develop insight into the contributions of both the distributional and CQL additions to the DeD framework by considering ablations to DistDeD where each component is removed. Additional details of all experiments are contained in Section A.3 in the Appendix where there can also be found further experimental analyses in Section A.4, such as the effects of learning with reduced data (see Section A.4.4).

---

[2]All code for data extraction and preprocessing as well as for defining and training DistDeD models can be found at `https://github.com/MLforHealth/DistDeD`.

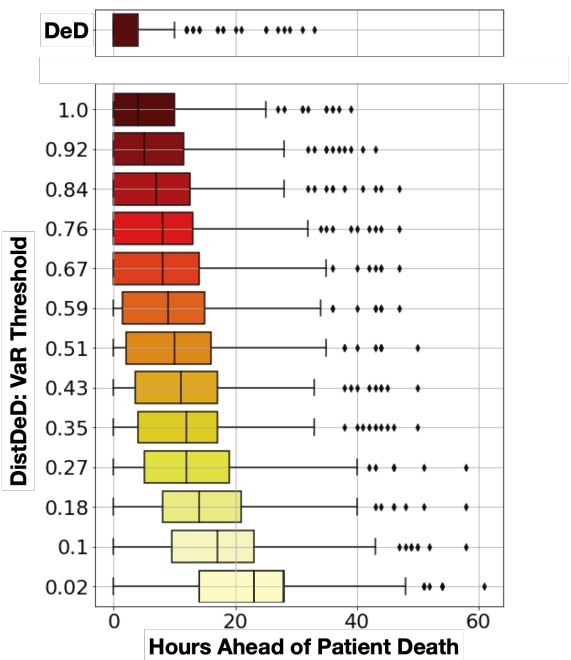

Figure 5: The number of hours before patient death that DistDeD and DeD raise warning.

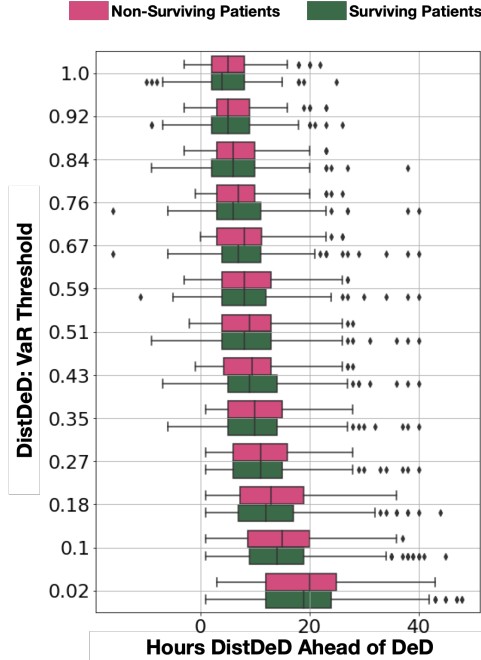

Figure 6: The number of hours that DistDeD detects patient deterioration and first raises a flag before DeD.

## 6.2   Results

As outlined in Section 6.1 we highlight the importance of accounting for risk when thinking about dead-ends and validate the following aspects of DistDeD. First, we assess the performance of DistDeD by demonstrating how DistDeD can provide an earlier indication of risk in comparison to other baselines and notably, outperforms DeD across all settings. Second, we demonstrate the utility of having a tunable assessment of risk that allows for domain experts to easily apply and adapt our method to different contexts, hospital settings and illnesses. Finally, we show that including a CQL penalty in the DistDeD framework further improves performance in comparison to other baselines.

### 6.2.1   DistDeD Provides Earlier Warning of Patient Risk

We assess the ability of DistDeD to provide an early warning of patient risk in comparison to the original medical dead-ends framework, DeD. Figure 5 shows for non-survivors, the number of hours ahead of death that DistDeD raises a warning flag and how this changes with varying choices of VaR. In comparison to DeD, DistDeD is able to raise flags much earlier warning of up to 25 hours in advance across all values of VaR, thereby enabling timely intervention in safety-critical settings.

To assess DistDeD's ability to raise flags in different contexts, we also compare how its performance varies across both surviving and non-surviving patients. These results are shown in Figure 6. In general we note that for both patient groups, DistDeD is able to detect patient deterioration and provide early warning of up to 20 hours in advance depending on the choice of VaR thresholds in comparison to DeD. The performance across both surviving and non-surviving patients is very similar.

### 6.2.2   DistDeD Allows for a Tunable Assessment of Risk

Note that because DistDeD explicitly uses the Value at Risk threshold parameter $\alpha$ to provide an assessment of risk, it can easily be adapted and tuned to various scenarios depending on how risk-averse a user would like to be. In addition, the choice of the thresholds $\delta_D$ and $\delta_R$ can be further adjusted to improve the precision of estimates of the risk of encountering a dead-end. For instance, in an ICU setting where timely intervention is crucial, a clinician may choose to adopt lower $\alpha$ and higher $\delta_D$ & $\delta_R$ threshold values to be more conservative such that flags may be raised earlier if necessary.

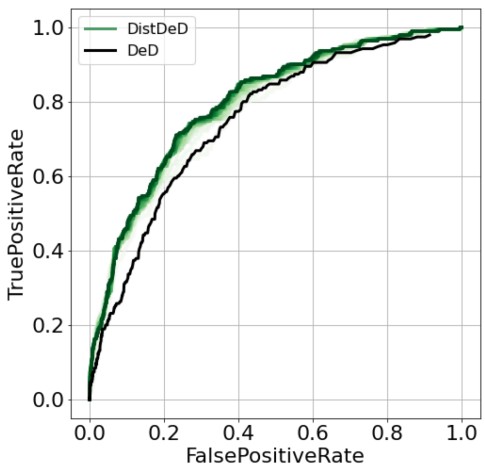
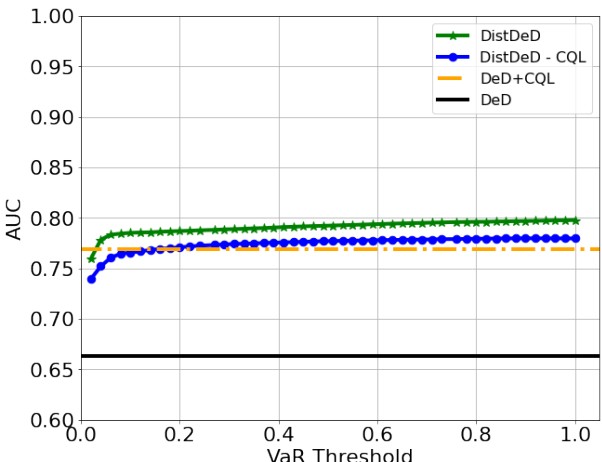

Figure 7: ROC curve comparison between CVaR$_\alpha$ settings of DistDeD (green) and DeD (black), demonstrating DistDeD's robust improvement over DeD.

Figure 8: Evaluation of the CQL penalty in terms of area under the ROC curve (AUC), comparing DistDeD, DeD and two ablations to DistDeD.

In our experiments, we evaluate DistDeD and DeD over all possible settings of $\delta_D$ and $\delta_R$ to assess the sensitivity of those settings when computing the True Positive Rate (TPR) and False Positive Rate (FPR) of determining patient risk. We also continue to evaluate DistDeD over a range of CVaR$_\alpha$ settings. Here, TPR corresponds to the percentage of non-survivor trajectories that are flagged, while FPR corresponds to the percentage of survivor trajectories that are flagged. Figure 7 shows a comparison of ROC curves derived from the DistDeD and DeD frameworks to exhibit how each balance the TPR and FPR tradeoff. For DistDeD we evaluated the TPR and FPR for a range of $\alpha$ values to identify whether there was a particular level of conservatism (or optimism) that would perform worse than DeD. However, we observe that DistDeD robustly outperforms DeD finding a higher TPR while having a low FPR in comparison, across all settings of $\alpha$, $\delta_D$ and $\delta_R$. Overall, having an *tunable* assessment of risk also enables a domain expert like a clinician balance the benefits of early warning with the risk of potential false positive indications of risk, where a patient at low-risk is potentially flagged. Moreover, a higher TPR counteracts an increased FPR when we are more conservative in the DistDeD framework.

### 6.2.3 CQL Enhances DistDeD Performance

In order to assess the individual contributions of implementing a distributional estimate of the risk of encountering a dead-end *and* constraining the values with CQL, we evaluate separate ablations to DistDeD by computing the area under the ROC curve derived from each approach. Figure 8 shows the performance comparison of DistDeD versus DeD and these two ablations that i) exclude a CQL penalty from the DistDeD framework and ii) incorporate a CQL penalty into the standard DeD framework. Overall, we see that the DistDeD framework outperforms the baselines in terms of AUC across varying levels of the VaR threshold. We summarize the findings with the maximum AUC of each approach in Table 1. In total, DistDeD (which combines the IQN and a CQL penalty) provides an average AUC of 0.7912 while DeD results in an AUC of 0.6629, resulting in as much as a 20% improvement in the precision of identifying dead-end states.

|  | Architecture | |
| --- | --- | --- |
|  | DDQN | IQN |
| No Penalty | 0.6629 | 0.7744 |
| CQL Penalty | 0.7687 | **0.7912** |

Table 1: Comparison of AUC when considering each improvement to DeD, 1) incorporating the CQL penalty and 2) modeling the full distributions of the expected return. Values represented here for the distributional components represent the mean value over all settings of VaR$_\alpha$.

# 7 Discussion

In this paper we have presented our justification, foundational evidence as well as our preliminary findings supporting the development of the DistDeD framework which incorporates a more complete notion of risk when identifying dead-ends in safety-critical scenarios. We do so by leveraging distributional dynamic programming to form estimates of the full return distribution from which we can calculate the expected worst-case outcome for each available action. This form of risk-estimation enables a more tangible decision surface for determining which actions to avoid and can be tuned according to the requirements or preferences set forward by human experts that may interact with the trained DistDeD models.

Our DistDeD approach is based around risk-sensitive estimates of the expected worst-case outcome and is thereby contributes a conservative decision support framework. This framework is well suited for complex safety-critical situations where learning is completed in a fully offline manner.

**Limitations** While DistDeD is a promising framework for decision support in safety-critical domains with limited offline data, there are certain core limitations. The techniques described in this paper have been explored in the context of discrete action spaces only. However in scenarios where continuous actions are featured, analyses with the DistDeD framework may have to be adapted to identify potential dead-ends. In addition, the method considers only cases where a binary reward signal is observed on the terminal state only. However, several applications may require us to account for intermediate and continuous outcomes as well. Moreover, the framework only explores a medical scenario where dead-ends are derived from a single condition whereas in reality, many concomitant conditions may exist, which contribute to and are associated with different dead-end regions. Finally, we do not make any causal claims about the impact of each action on the outcomes of interest. Future work may explore how to address some of these issues. In addition, we are currently in the process of applying DistDeD to real-world healthcare challenges in partnership with clinicians to further demonstrate its utility in that setting. We do however anticipate that DistDeD is widely useful for all safety-critical domains that may beset with limited offline data.

### Broader Impact

This work serves as a proof of concept for identifying regions of risk in safety-critical settings, learning from offline data. While promising, it has not been thoroughly validated for immediate use in real environments. Despite the demonstrated utility of the DistDeD framework in healthcare problems, it should never be used in isolation to exclude patients from being treated, e.g., not admitting patients or blindly ignore treatments. The risk identification aspect of DistDeD demonstrated in this paper is to signal impending high-risk situations early enough so that the human decision maker has time to correct the course of action. This may help experts make better decisions and avoid circumstances that may lead to irrecoverably negative outcomes. The intention of our approach is to assist domain experts by highlighting possibly unanticipated risks when making decisions and is not to be used as a stand-alone tool nor as a replacement of a human operator. Misuse of this algorithmic solution could carry significant risk to the well-being and survival of critical systems and individuals placed in the care of the expert.

The primary goal of this work is to improve upon the established DeD proof of concept, where high-risk situations can be avoided in context of a system's state (Fatemi et al., 2021). We present a distributional estimate of this risk profile which enables earlier detection of possible dead-ends as well as facilitating a tunable framework for adaptation to each individual task. In acute care scenarios, all decisions come with inherent risk profiles and potential harms. In this spirit, we endeavor to provide a flexible tool for clinical experts to gain an earlier indication when specific decisions or their patient's health state may carry a measure of outstanding risk.

### Author Contributions

TK and SP conceived and designed the research questions as well as wrote the paper. TK extracted and processed the data, designed and executed the experiments, and performed the analyses. MG provided input on possible uses of the proposed framework in clinical settings, provided funding, and reviewed the paper prior to it being made public.

**Acknowledgments**

We thank our many colleagues and friends who contributed to thoughtful discussions and provided timely advice to improve this work. Specifically, we appreciate the encouragement and enthusiasm provided by Vinith Suriyakumar, Haoran Zhang, Mehdi Fatemi, Will Dabney and Marc Bellemare. We are grateful for the feedback provided by Swami Sankaranarayanan, Qixuan Jin, Tom Hartvigsen, Intae Moon and the anonymous reviewers who helped improve the writing of the paper.

This research was supported in part by Microsoft Research, a CIFAR AI Chair at the Vector Institute, a Canada Research Council Chair, and an NSERC Discovery Grant.

Resources used in preparing this research were provided, in part, by the Province of Ontario, the Government of Canada through CIFAR, and companies sponsoring the Vector Institute `www.vectorinstitute.ai/#partners`.

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

# A    Appendix

## A.1    Sepsis Patient Cohort Details

We use the MIMIC-IV (Medical Information Mart for Intensive Care; v2.0) database, sourced from the Beth Israel Deaconess Medical Center in Boston, Massachusetts Johnson et al. (2020). This database contains deidentified treatment records of patients admitted to critical care units (CCU, CSRU, MICU, SICU, TSICU). The database includes data collected from 76,540 distinct hospital admissions of patients over 16 years of age for a period of 12 years from 2008 to 2019 (inclusive). The MIMIC database has been used in many reinforcement learning for health care projects, including mechanical ventilation and sepsis treatment problems. There are various preprocessing steps that are performed on the MIMIC-IV database in order to obtain the cohort of patients and their relevant observables for sepsis cohort used in this study.

To extract and process the data, we follow the approach used in (Fatemi et al., 2021). This includes all ICU patients over 18 years of age who have some presumed onset of sepsis (following the Sepsis 3 criterion) during their initial encounter in the ICU after admission, with a duration of at least 12 hours. We limited the observation window of each patient encounter from at most 24 hours before to at most 48 hours after presumed sepsis onset. We also constrained collection to include only those patients admitted to the Medical ICU (MICU) on that initial encounter. These criteria provide a cohort of 6,188 patients, among which there is an observed mortality rate of 13.5%, where mortality is determined by patient expiration within 48h of the final observation. Observations are processed and aggregated into hourly windows with treatment decisions (administering fluids, vasopressors, or both) discretized into 5 volumetric categories. All data is normalized to zero-mean and unit variance and missing values are zero-imputed with an binary mask appended to indicate which features were observed at each timestep. We report the 42 features used in the construction of this patient cohort in Table 2 with high-level statistics in Table 3.

## A.2    DistDeD Architecture Details

In this section, we outline the motivation, design, and training of the various architectures used to formulate the DistDeD framework. To account for the irregularity and temporal dependence of the observations made

Table 2: Patient features used for learning state representations for predicting future observations

| Age | Gender | Weight (kg) | Height |
|---|---|---|---|
| Heart Rate | Sys. BP | Dia. BP | Mean BP |
| Respiratory Rate | Body Temp (C) | Glucose | SO2 |
| PaO2 | PaCO2 | FiO2 | PaO2 / FiO2 |
| Arterial pH | Base Excess | Chloride | Calcium |
| Potassium | Sodium | Lactate | Hematocrit |
| Hemoglobin | Platelet | White Blood Cells | Albumin |
| Anion Gap | Bicarbonate (HCO3) | PT | PTT |
| Glascow Coma Scale | SpO2 | BUN | Creatinine |
| INR | Bilirubin | SGOT (AST) | SGPT (ALT) |
| Urine Output | Mech. Ventilation | | |

with the medical data, we encode the data using an online Neural Controlled Differential Equation (NCDE). This provides a fixed dimensional state representation at each time step (here, aggregated by hour), to align with the frequency of treatment decisions. The encoded state representations are then used as input into the independent implicit quantile networks (IQN), used to represent the D- and R- Networks to estimate the risk of encountering a dead-end.

**Training** We train the NCDE for state construction as well as the IQN instantiations for the D-, and R- Networks in an offline manner. All models are trained with 75% of the data (4,014 surviving patients, 627 patients who died),validated with 5% (268 survivors, 42 nonsurvivors), and we report all results on the remaining held out 20% (1,070 survivors, 167 nonsurvivors). In order to account for the data imbalance between positive and negative outcomes, we follow a similar training procedure as DeD (Fatemi et al., 2021) where every sampled minibatch is ensured to contain a proportion of terminal transitions from non-surviving patient trajectories. This amplifies the training for conditions that lead to negative outcomes, ensuring that the D- and R- Networks are able to recognize scenarios that carry risk of encountering dead-ends.[3]

### A.2.1 Neural Controlled Differential Equation

Neural Differential Equations (NDEs; Chen et al. (2018)) have become a popular modeling framework for handling complex temporal data due to their flexibility and the ability to model data in continuous time. In particular, they are well matched for irregularly sampled (e.g. partially observed) data such as is common in healthcare. NDEs learn a continuous latent representation of the dynamics underlying the observed data in a fixed dimension representation; adapting for missingness, various periodic frequencies among features, as well as complex interactions between features (Kidger, 2022). These reasons provide a distinct motivation for using NDEs for processing fixed representations of healthcare data.

In this work we use a variant of NDE, Neural Controlled Differential Equations (NCDE; Morrill et al. (2021)). NCDEs are designed to process irregular time series with a latent process that affects the evolution of the observed features. The particular variant of NCDE we use is designed to operate in online settings, only incorporating historical information to encode representations of the current time step. This is a departure from standard NDE methods that execute a forward-backward time of autoregression when representing the latent dynamics of the time series. By restricting ourselves to online types of processing, we honor the reality with which data is received in a healthcare setting which leads to a more realistic implementation. Otherwise, we may risk biasing the inference over missing data intervals using future observations. For specific algorithmic details of the NCDE, we refer the reader to Morrill et al. (2021).

For our purposes, we train the NCDE as a continuous time autoencoder of the irregular patient observations. This provides a fixed dimension representation of a patient's state at hourly intervals, to match with the frequency of treatment decisions in our extracted data. Following procedures set forth by Morrill et al. (2021), we lightly pre-process the data with rectilinear interpolation (where each patient trajectory is han-

---

[3]All code for data extraction and preprocessing as well as for defining and training DistDeD models can be found at `https://github.com/MLforHealth/DistDeD`.

Table 3: MIMIC-IV Sepsis Cohort Statistics: `Median` (25% - 75% quantiles)

| Variable | MIMIC ($n = 6188$) | Variable | MIMIC ($n = 6,188$) |
|---|---|---|---|
| **Demographics** | | **Outcomes** | |
| Age, years | 68.0 (57.0.-80.0) | Deceased | 836 (13.51%) |
| Age range, years | | Vasopressors administered | 2241 (36.2%) |
|    18-29 | 171 (2.8%) | Fluids administered | 6032 (97.5%) |
|    30-39 | 269 (4.3%) | Ventilator used | 2201 (35.6%) |
|    40-49 | 414 (6.7%) | | |
|    50-59 | 911 (14.7%) | | |
|    60-69 | 1426 (23%) | | |
|    70-79 | 1332 (21.5%) | | |
|    80-89 | 1207 (19.5%) | | |
|    $\geq$90 | 458 (7.4%) | | |
| Gender | | | |
|    Male | 3251 (52.54%) | | |
|    Female | 2937 (47.46%) | | |
| **Physical exam findings** | | | |
| Temperature (°C) | 36.8 (36.6-37.3) | | |
| Weight (kg) | 75.7 (63.1-91.0) | | |
| Height (cm) | 168.0 (160.0-177.0) | | |
| Heart rate (beats per minute) | 88.0 (76.0-103.0) | | |
| Respiratory rate (breaths per minute) | 20.0 (16.0-24.0) | | |
| Systolic blood pressure (mmHg) | 112.0 (100.00-127.0) | | |
| Diastolic blood pressure (mmHg) | 61.0 (52.0-70.0) | | |
| Mean arterial pressure (mmHg) | 75.8 (60.8-90.8) | | |
| Fraction of inspired oxygen (%) | 74.0 (66.0-84.0) | | |
| P/F ratio | 165.0 (104.2-258.0) | | |
| Glasgow Coma Scale | 15.0 (14.0-15.0) | | |
| **Laboratory findings** | | | |
| Hemotology | | Coagulation | |
|    White blood cells (thousands/$\mu$L) | 10.9 (7.0-16.4) |    Prothrombin time (sec) | 15.1 (13.3-18.5) |
|    Platelets (thousands/$\mu$L) | 160.0 (97.0-240.0) |    Partial thromboplastin time (sec) | 33.2 (28.5-41.8) |
|    Hemoglobin (mg/dL) | 9.5 (8.2-10.9) |    INR | 1.4 (1.2-1.7) |
|    Hematocrit (mg/dL) | 28.8 (25.2-33.0) | | |
|    Base Excess (mmol/L) | -1.0 (-6.0-1.0) | | |
| Chemistry | | Blood gas | |
|    Sodium (mmol/L) | 136.0 (132.0-140.0) |    Arterial pH | 7.3 (7.2-7.4) |
|    Potassium (mmol/L) | 4.2 (3.7-4.9) |    Oxygen saturation (%) | 92.0 (74.0-97.0) |
|    Calcium (mg/L) | 1.1 (0.9-1.2) |    SpO2 (%) | 97.0 (95.0-99.0) |
|    Chloride (mmol/L) | 106.0 (102.0-111.1) |    Partial pressure of O2 (mmHg) | 77.0 (48.0-111.0) |
|    Bicarbonate (mmol/L) | 22.0 (19.0-25.0) |    Partial pressure of CO2 (mmHg) | 42.0 (35.0-50.0) |
|    Blood urea nitrogen (mg/dL) | 25.0 (15.0-43.0) | | |
|    Creatinine (mg/dL) | 1.1 (0.7-1.9) | | |
|    Albumin (mg/dL) | 2.8 (2.4-3.2) | | |
|    Anion Gap (mmol/L) | 14.0 (12.0-17.0) | | |
|    Glucose (mg/dL) | 138.0 (106.0-189.0) | | |
|    SGOT (units/L) | 52.0 (27.0-122.0) | | |
|    SGPT (units/L) | 39.0 (19.0-102.0) | | |
|    Lactate (mg/L) | 1.9 (1.3-3.3) | | |
|    Total bilirubin (mg/L) | 0.9 (0.5-2.5) | | |

dled independently) so as to signal when and where missing features occur. The NCDE is built around learning representations from an internal projection function (represented by a neural network), optimized using a differential equation solver. We fine-tuned the hyperparameters of this encoding function (most importantly the output embedding dimension) as well as optimization using `Ax` (Bakshy et al., 2018), an adaptive experimentation platform built on top of the `BoTorch` Bayesian Optimization library (Balandat et al., 2020).

When training the NCDE, we found that using a fixed learning rate with the Adam optimizer (Kingma & Ba, 2014) performed best. After 100 Bayesian Optimization trials, we found the following hyperparameter settings to provide the best performing NCDE model. For the encoding neural network, we used 2 layers with 80 hidden units in each with ReLU activations. The output dimension of this encoding network was

55, which provided the state representations then used as input to the Reinforcement Learning models. For optimization, the best learning rate was $5e-4$ over 30 epochs.

### A.2.2 DistDeD Value Functions

As outlined in Section 4, we construct two MDPs for estimating the return for negative and positive outcomes independently. With these independent learning objectives, we construct two independent value estimators based on Implicit Q-Networks (Dabney et al., 2018). For specific details on the development and training of these architectures, we refer the reader to the source literature. In summary, quantiles of the approximated distribution of return are approximated with sampled particles from a uniform distribution, which are then transformed with a learned projection function (represented by a neural network) to construct the implicit distribution. When training, a separate copy of the network parameters are kept as a target network to ensure more stable updates (following the double DQN strategy (Hasselt et al., 2016)), a number of samples $K$ are drawn from each distribution (the one we're optimizing and the target distribution), then using a Wasserstein metric the projected particles are brought closer together. To constrain the value estimates of this distribution from overestimation for actions not in the dataset, we include a CQL penalty (Kumar et al., 2020) following Ma et al. (2021). There is a trade-off between maximizing the fit of the value distributions and the strength of the CQL regularization. We can modulate this by including a multiplicative weight $\beta$ to the CQL penalty, which we treat as an additional hyperparameter.

As done when training the NCDE state constructor, we optimized the hyperparameters of the IQN models used to represent $Z_D$ and $Z_R$ in DistDeD, the CQL penalty weight, and optimization parameters using the `BoTorch` Bayesian Optimization library Balandat et al. (2020) through the `Ax` API (Bakshy et al., 2018). The best performing hyperparameters used to define the IQN and CQL penalty were chosen after running 100 optimization trials. For the IQN, the projection neural network accepted a 55 dimensional input (from the NCDE), consisted of 2 layers with 16 hidden units in each, using ReLU activations. The number of samples K drawn each optimization step was set to 64. The target network parameters were updated after every 5 optimization steps using an exponentially-weighted moving average with parameter $\tau$ set to 0.005. By construction, the discount rate $\gamma$ is set to 1. For the weighting of the CQL penalty, $\beta = 0.035$. For optimization, we used Adam (Kingma & Ba, 2014) with the best performing learning rate found to be $2e-5$ over 75 epochs of training.

Of special note, the IQN architecture admits a family of risk-sensitive policies by constraining the space from which samples are drawn for the approximating distribution. The way the sampling space is constrained is called a distortion risk measure which influences the underlying value distributions to be more risk seeking or averse. By design, these distortion measures can be selected to influence the estimation of the implicit distributions over expected return. However, we chose not to bias the estimation of these distributions since we are deploying the IQN in an offline setting and cannot recover from a poor modeling choice through the acquisition of new data from the environment (following the optimism under uncertainty principle that guides much of RL). We therefore do not employ any distortion measures, evaluating the CVaR in a post-hoc manner so as to maximize the utility of the underlying distribution to represent the observed data.

**Computing the CVaR**   To compute the CVaR of $Z_D$ and $Z_R$, we evaluate the fully trained IQN models using held out test data. The full distributions are then sampled using $K_{\text{test}} = 1000$. We then sort the resultant estimated values for each particle and select the fraction of smallest values corresponding the the chosen value of $\alpha$. For example, if $\alpha = 0.35$, we would then take the smallest 350 values of the 1000 samples. The $\text{VaR}_{\alpha=0.35}$ would be the maximum value of this subsampled portion of the estimated distribution. Then, the $\text{CVaR}_{\alpha=0.35}$ would be the average value of the 350 sample subset. We use this quantity then to compare with the DistDeD thresholds $\delta_D$ and $\delta_R$ to determine whether or not an action should be avoided or whether the state is a dead-end as outlined in Section 4.

### A.3  Details of experimental setup

This section lays out the details of the models used in all experiments as well as the relevant settings of each experiment presented in Section 5 and Section 6. We start by listing the important hyperparameters

for both DeD and DistDeD with a description of their function in Table 4. Many of the components of each approach share the same description. We follow this description with a description of each experiment, listing relevant settings and parameters for the models used as well as analyses performed.

Table 4: Listing of parameters and hyperparameters for dead-end discovery

| DeD (Fatemi et al., 2021) | | DistDeD (this work) | |
|---|---|---|---|
| D-Network | Neural Network used to estimate the value of treatment options in relation to a negative terminal outcome | | |
| $Q_D(s,a)$ | DDQN | $Z_D(s,a)$ | IQN |
| R-Network | Neural Network used to estimate the value of treatment options in relation to a positive terminal outcome | | |
| $Q_R(s,a)$ | DDQN | $Z_R(s,a)$ | IQN |
| $\gamma$ | Discount rate for training value functions, $\gamma = 1$ always | | |
| $\delta_D$ | Threshold used to determine when to flag the values produced by the D-Network | | |
| $\delta_R$ | Threshold used to determine when to flag the values produced by the R-Network | | |
| $R_D(s,a)$ | Reward function used in $\mathcal{M}_D$ to train the D-Network. All positive terminal states have reward of 0, while negative terminal states have reward of -1. All other states receive a reward of 0. | | |
| $R_R(s,a)$ | Reward function used in $\mathcal{M}_R$ to train the R-Network. All positive terminal states have reward of +1, while negative terminal states have reward of 0. All other states receive a reward of 0. | | |
| | | $N, K$ | the number of samples drawn from the learned quantile function, differs between training ($N$) and evaluation ($K$) |
| | | $\beta$ | the weight given to the CQL penalty during optimization |

### A.3.1 Experimental Details for the Illustrative Demonstration

In this subsection, we'll provide some additional details about the toy domain LifeGate which was introduced by Fatemi et al. (2021) and the empirical analyses done to compare DeD and DistDeD (all relevant parameters are contained in Table 5). In this domain, an agent is tasked with navigating to a goal region by making it's way around a barrier, while learning to avoid a "dead-end zone" at the right side of the barrier. In this zone, no matter what actions the agent takes it will be pushed to the right toward the negative terminal region after a random number of steps. Even in this simple domain, we found that prior dead-end discovery approaches (DeD) would overestimate the safety of actions around this dead-end zone (see Figure 3) and only raise a flag about the risk of an agent reaching a dead-end once it was squarely within this zone (see Figure 4).

Using this toy domain, we set out to visualize the learned value distributions provided with DistDeD as an informative means to demonstrate it's utility broadly. To do so, we collected 1 million transitions using a random policy with the agent being initialized randomly all over the environment. With this data, we were

able to then train the value functions for DeD and DistDeD using the DDQN (Hasselt et al., 2016) and IQN (Dabney et al., 2018) algorithms, respectively. Additionally, we applied a CQL penalty (Kumar et al., 2020) to the IQN training of DistDeD. Using the published $\delta_D$ and $\delta_R$ thresholds published by Fatemi et al. (2021) for DeD and those empirically derived for DistDeD (see Section A.4.1), which are included in Table 5, we could then quantitatively compare the performance of these two approaches in LifeGate.

In the first experimental comparison (see Figure 3), we evaluated a large selection of states in the LifeGate domain choosing two to visualize the outputs of the value functions. For DeD, we simply recorded the value estimate provided by the D- and R- Networks. With DistDeD, we sampled 1000 points from the underlying quantile functions used to approximate the return distributions within the IQN from both the D- and R- Network. This allowed us to construct representative value distributions for each action. With each distribution, and a chosen $\alpha$ value for calculating the value-at-risk (VaR) and thereby the conditional value-at-risk (CVaR) we could then visualize what the estimated "worst-case value" of each action was.

In the second and third experimental comparison using the LifeGate domain (see Figure 4), we evaluated three hand-designed policies used to collect trajectories. Two of these policies were purposefully made to be suboptimal (meaning that they would traverse through the dead-end zone) to demonstrate how early DistDeD would identify the risk of reaching a dead-end. We quantified this advantage by collection ten-thousand additional trajectories, following these suboptimal policies, and recording the time either $Q_D$ and $Q_R$ or CVaR($Z_D$) and CVaR($Z_R$) violated their associated thresholds. We then subtracted the time the agent reached the dead-end zone from this recorded time. Using the 10,000 collected trajectories, we could then aggregate statistics about the time differential and how much earlier DistDeD signaled risk when compared to DeD.

Table 5: Experimental parameters for LifeGate

| DeD (Fatemi et al., 2021) | | DistDeD (this work) | |
|---|---|---|---|
| $Q_D(s,a)$ | DDQN 2 layers of 32 units | $Z_D(s,a)$ | IQN 2 layers of 32 units |
| $Q_R(s,a)$ | DDQN 2 layers of 32 units | $Z_R(s,a)$ | IQN 2 layers of 32 units |
| $\gamma = 1$ | | $\gamma = 1$ | |
| $\delta_D$ | -0.15 | $\delta_D$ | -0.5 |
| $\delta_R$ | 0.85 | $\delta_R$ | 0.5 |
| | | $N$ | 8 |
| | | $K$ | 1000 |
| | | $\alpha$ | 0.1 |
| | | $\beta$ | 0.1 |

| | |
|---|---|
| # of datapoints | $1e^6$ |
| # of training epochs | 50 |
| learning rate | $1e^{-3}$ |
| # of evaluation trajectories | 10,000 |
| dimension of state space $\mathcal{S}$ | 2 |
| dimension of action space $\mathcal{A}$ | 5 |

### A.3.2 Experimental Details for Sepsis Treatment Evaluation

All medical data used in this paper is derived from the MIMIC-IV database, as described in Section A.1. After filtering and data exploration, we ended up with 6,188 high quality trajectories of patients who developed Sepsis and were admitted to the intensive care unit. Approximately 13.5% of the trajectories end in the patient dying, reaching our defined negative terminal condition. More detailed statistics about the patient cohort can be found in Table 3.

Since observations derived from electronic health records are irregular and sparse, we follow the previous literature applying RL to healthcare and learn a fixed-dimensional latent encoding of the data over time (Killian et al., 2020). We chose to model this encoder with a Neural Controlled Differential Equation (NCDE) (Morrill et al., 2021), trained via an objective to reconstruct the currently provided observation. Details about training the NCDE are given in Section A.2.1. Our best performing NCDE model took the 42 dimensional observations and projected them into a 55 dimensional latent state space, which was then used to train the D- and R- Networks for the value functions underlying DeD and DistDeD, details of which can be found in Section A.2.2. A table summarizing high-level parameters about the imposed MDP used to define the experiments in Section 6 can be found in Table 6.

In Section 6.2.1, we determined a single set of thresholds for DistDeD following the same analysis done by Fatemi et al. (2021). We highlight how this is done in Section A.4.1. In essence, we plot sets of histograms (one for surviving patients, one for nonsurviving patients) of the computed CVaR for both the D- and R-Networks over all states for each time step, for all $\alpha$ values. We then attempted to select $\delta_D$ and $\delta_R$ that would separate the nonsurviving patient values from the surviving patient values, minimizing as many false positives (values from surviving patients that fall below the thresholds). Using these thresholds (for both DeD and DistDeD), we determine the first time step when the median of the CVaR values over all actions fell below the thresholds, for both D- and R-Networks respectively, which signifies a significant risk of the patient reaching a dead-end. In Figure 5, we measure how far ahead of patient death, in the case of non-surviving patients, this first flag is raised. The box plots are taken over the setting of CVaR risk threshold $\alpha$. Figure 6 represents the spirit of our empirical analyses and and we compare the temporal difference between DistDeD and DeD directly for all patients. We see again that DistDeD provides earlier indication of patient health deterioration, particularly as lower values of $\alpha$ are selected. This analysis introduced an important question about the balance between early warning and increased false positives.

We address the concern of increased false positives in Section 6.2.2 by defining the notions of true and false positive dead-end discovery and also investigate the range of performance acheived by selecting different values for the thresholds $\delta_D$ and $\delta_R$. This also enabled us to establish a receiver operating characteristic (ROC) as a metric to holistically evaluate the performance of DistDeD vs. DeD. We evaluated each patient trajectory and aggregated the rate of nonsurviving patients having been correctly flagged by either DeD or DistDeD as well as the rate of surviving patients "wrongly" flagged. We construct the ROC curve by evaluating the sensitivity of the dead-end discovery process in each DistDeD and DeD by varying the thresholds $\delta_D$ and $\delta_R$ over 100 possible settings to provide a more complete picture of the performance of any method. Figure 7 shows this comparison, allowing us to see that DistDeD robustly outperforms DeD, regardless of the choice of $\alpha$ (each green curve corresponds to an independent setting of $\alpha$).

In Section 6.2.3, we repeat all of the above training and evaluation paradigms for two ablations of DistDeD by removing either the distributional component (essentially running DeD with a CQL penalty, which could be thought of as a separate baseline) or the CQL penalty. We present in Figure 8 the summary of this evaluation using the quantitative measure of Area under the ROC curve as a comparison. Table 1 takes the maximum AUC of each approach (picking the best configuration of $\alpha$ for DistDeD and DistDeD without CQL). Here, we conclude that DistDeD provides a 20% improvement over DeD using this AUC metric.

## A.4   Additional Experimental Analysis

### A.4.1   Preliminary selection of decision thresholds

In Figure 9 we present a visual summary of how the thresholds $\delta_D$ and $\delta_R$ are empirically determined. Conceptually, we want to select thresholds that minimize the number of "false positives" that occur, meaning we don't want to unnecessarily flag trajectories arising from patients who ultimately survived. We plot the histograms of the assessed values for both non-surviving (blue) and surviving (green) patients for both the D-Network (top) and R-Network (bottom) using the validation set. To visualize how the estimated values change throughout the recorded trajectory (max 72 hours before termination) we also look at successive time periods when plotting the histograms. Unsurprisingly, as the trajectories near termination, the states from non-surviving patients have lower estimated value (being near to death). The choice of threshold is made

Table 6: Experimental parameters for Sepsis Treatment Experiments

| DeD (Fatemi et al., 2021) | | DistDeD (this work) | |
|---|---|---|---|
| $Q_D(s,a)$ | DDQN | $Z_D(s,a)$ | IQN |
| | 2 layers of 64 units | | 2 layers of 16 units |
| $Q_R(s,a)$ | DDQN | $Z_R(s,a)$ | IQN |
| | 2 layers of 64 units | | 2 layers of 16 units |
| Training epochs | 100 | Training epochs | 75 |
| $\gamma = 1$ | | $\gamma = 1$ | |
| $\delta_D$ | -0.15 | $\delta_D$ | Experiment dependent |
| $\delta_R$ | 0.85 | $\delta_R$ | Experiment Dependent |
| learning rate | $1e^{-4}$ | learning rate | $2e^{-5}$ |
| | | $N$ | 64 |
| | | $K$ | 1000 |
| | | $\alpha$ | 50 settings linearly spaced along [0,1] |
| | | $\beta$ | 0.035 |

| **NCDE Properties** | |
|---|---|
| Number of training epochs | 30 |
| Encoder Neural Network | 2 layers with 80 hidden units |
| input dimension | 42 |
| output dimension | 55 |
| learning rate | $5e^{-4}$ |

| **General MDP Properties** | |
|---|---|
| # of patient trajectories | $6,188$ |
| minimum trajectory length | 12 |
| median trajectory length | 42 |
| maximum trajectory length | 72 |
| # of features | 42 |
| dimension of state space $\mathcal{S}$ | 55 |
| dimension of action space $\mathcal{A}$ | 25 |

| **Experiments in Section 6.2.1** | | | |
|---|---|---|---|
| $\delta_D$ | -0.15 | $\delta_D$ | -0.5 |
| $\delta_R$ | 0.85 | $\delta_R$ | 0.5 |

| **Experiments in Section 6.2.2** | | | |
|---|---|---|---|
| $\delta_D$ | 100 settings w/in [-1,0] | $\delta_D$ | 100 settings w/in [-1,0] |
| $\delta_R$ | $1+\delta_D$ | $\delta_R$ | $1+\delta_D$ |

to provide as early of an separation of the estimated values between non-surviving and surviving patient as possible.

While this approach carries some precedence, as it follows that done by Fatemi et al. (2021), but it's clear how tedious and in-exact this process is. This is what led to the analysis provided in Section 6.2.2, where we evaluated all possible settings of the thresholds when constructing the ROC curves in Figure 7. By providing the full information of possible precision and anticipated risk of false-positives, we enable the human expert to tune DistDeD according to the characteristics of the task. We suggest that this is a far superior approach to selecting the $\delta_D$ and $\delta_R$.

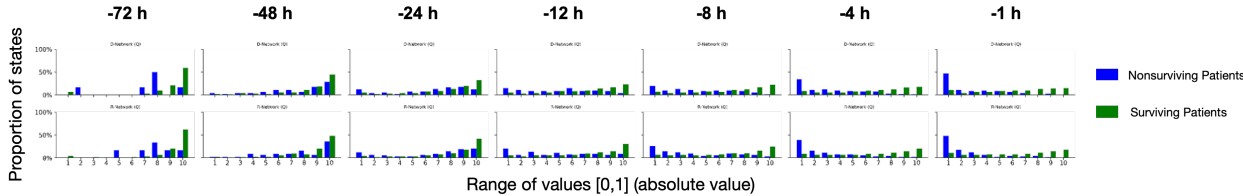

Figure 9: The evolution of estimated values using DistDeD over the course of the recorded patient trajectories (72 hours in total prior to termination), represented as histograms. The top row corresponds to the D-Network while the bottom row is derived from the R-Network. Estimated values from non-surviving patients are plotted in blue while those from surviving patients are plotted in green.

| | Non-Surviving Patients | | Surviving Patients | |
|---|---|---|---|---|
| $\alpha$ | DeD | DistDeD | DeD | DistDeD |
| 0.05 | 59.281 | 4.790 | 88.982 | 20.074 |
| 0.1 | 59.281 | 8.982 | 88.982 | 30.439 |
| 0.15 | 59.281 | 10.180 | 88.982 | 37.721 |
| 0.2 | 59.281 | 12.574 | 88.982 | 44.071 |
| 0.25 | 59.281 | 13.772 | 88.982 | 49.393 |
| 0.3 | 59.281 | 15.569 | 88.982 | 54.435 |
| 0.35 | 59.281 | 17.964 | 88.982 | 57.330 |
| 0.4 | 59.281 | 19.760 | 88.982 | 60.598 |
| 0.45 | 59.281 | 21.557 | 88.982 | 63.772 |
| 0.5 | 59.281 | 22.754 | 88.982 | 66.013 |
| 0.55 | 59.281 | 23.952 | 88.982 | 68.627 |
| 0.6 | 59.281 | 26.347 | 88.982 | 70.588 |
| 0.65 | 59.281 | 28.144 | 88.982 | 72.549 |
| 0.7 | 59.281 | 30.539 | 88.982 | 74.510 |
| 0.75 | 59.281 | 31.737 | 88.982 | 76.657 |
| 0.8 | 59.281 | 34.131 | 88.982 | 78.711 |
| 0.85 | 59.281 | 35.329 | 88.982 | 79.925 |
| 0.9 | 59.281 | 36.527 | 88.982 | 80.486 |
| 0.95 | 59.281 | 38.323 | 88.982 | 81.979 |
| 1.0 | 59.281 | 41.916 | 88.982 | 83.660 |

Table 7: Percentage of Patient Trajectories Missed. For DistDeD, $\delta_D = -0.5$, $\delta_R = 0.5$

### A.4.2 DistDeD recovers risky trajectories overlooked by DeD

While confirming the analysis underlying the results presented in Section 6.2.1, we were surprised to find that a significant number of non-surviving patient trajectories went undetected by DeD. In fact, nearly 60% of this high-risk subpopulation registered no indication from the prior dead-end discovery method. In Table 7 we present the proportion of trajectories (both non-suriving and surviving) where a flag is not raised for their duration, comparing between DeD and DistDeD. We also evaluated a range of $\alpha$ values used to calculate the CVaR of the estimated return distribution. We see that as $\alpha$ decreases, corresponding to a more conservative estimation of risk, that fewer non-surviving patient trajectories are missed at a cost of flagging more surviving patients. In concert with the results presented in Section 6.2.2, this table helps characterize the trade-off with early warning and the number of "false positives" that DistDeD provides. By providing this full range of options as an immediate consequence of the design of DistDeD, we empower the human decision maker to select the best setting of our proposed framework for their use-case.

### A.4.3 DistDeD performance suffers through an increase in conservatism

By construction, DistDeD is more conservative than prior dead-end discovery approaches. This is achieved in two ways: *a)* the choice of value at risk threshold ($\alpha$) and *b)* the weight ($\beta$) that the CQL penalty is given when optimizing the D- and R- Networks. We have demonstrated the tradeoff between high-levels of conservatism and performance by choosing a small value for the value-at-risk $\alpha$ in Figures 5,6,8 and Table 7. However the choice of $\beta$, which constrains value function learning, has a more significant impact on how much the value function is maximized with each gradient step. Increasing $\beta$ increases the conservatism of the learning algorithm, increasing the gap between the constrained and true value functions, as described in Section 3. In all experiments presented in Sections 5 and 6, we treated $\beta$ as a hyperparameter, tuned with the validation subset of our data. We leave the choice of $\alpha$ as a tunable parameter for the expert when evaluating the inferred value distributions, as described in Section A.2.2.

However, to demonstrate the effect of increased conservatism on the performance of DistDeD we investigated the effect of setting $\beta$ to larger values than those found through hyperparameter tuning. Specifically, we set $\beta = \{0.1, 0.2, 0.3, 0.4\}$ and compare to the optimal DistDeD performance (with $\beta = 0.35$), DistDeD without the CQL penalty, and DeD in Figure 10. With increased values for $\beta$, this is a significant reduction in DistDeD performance where only a subset of VaR thresholds surpass the performance of DeD. In fact, when $\beta$ in greater than 0.2, DistDeD wholly underperforms DeD in identifying dead-ends. Additionally, when using larger values for $\beta$, the effect of choosing $\alpha$ for the value-at-risk is more pronounced as there is a wider range of performance as $\alpha$ varies when $\beta$ is fixed.

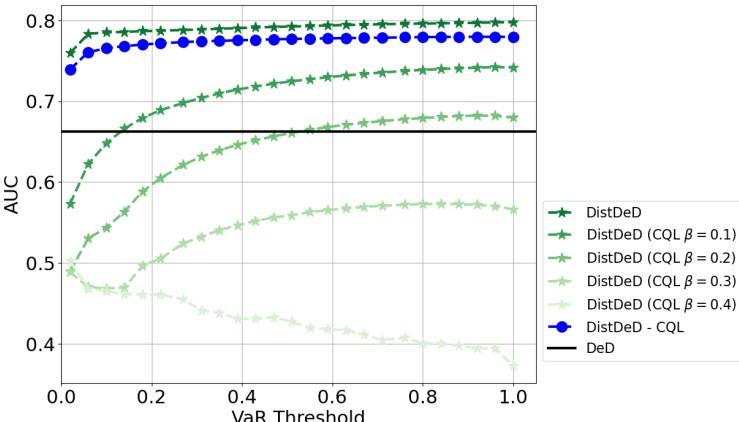

Figure 10: Demonstrating the effect of increased conservatism (increasing the CQL penalty weighting $\beta$) on DistDeD performance. Increasing the CQL weight serves to reduce the expressivity of the learned value distribution, constraining it further away from the true distribution. This corresponds to a reduction in performance of identifying dead-ends in the Septic patient population under consideration in this paper.

### A.4.4 DistDeD improves over DeD even in limited data settings

The use of a more complex object to represent the value return in DistDeD raises natural questions about how performance degrades in low data regimes. While distributional RL has been shown to be robust to such reductions in training data (Agarwal et al., 2020; Kumar et al., 2020), we evaluate DistDeD in comparison to DeD over a random subsampling of the training data. We ensure that the same proportion of positive to negative trajectories (e.g. derived from surviving and nonsurviving patients) is maintained when randomly sampling $\{10\%, 25\%, 50\%, 75\%\}$ subsets of the training data. We then retrain the D- and R- Networks for both DeD and DistDeD with each subset and evaluate the trained networks using the same procedure and test dataset as used in Section 6.

As demonstrated in Table 8 and Figure 11, we naturally see a reduction in test dead-end identification AUC. Yes, DistDeD's reduction in performance is not as sharp as DeD's while maintaining superiority over the prior approach. This confirms the findings in prior literature investigating the effect of low data regimes

on the performance of distributional RL algorithms. While low data regimes are shown to affect top-line performance across all learning algorithms, the effect is not disproportionately seen among distributional RL algorithms.

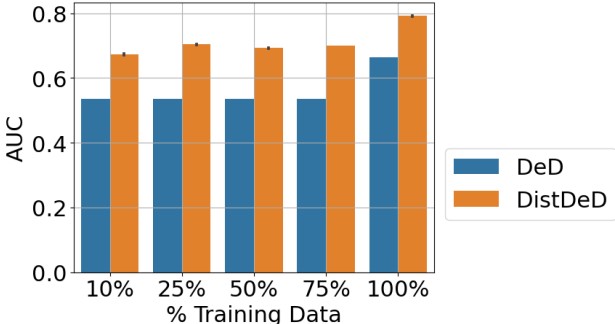

Figure 11: An investigation of the performance reduction of DistDeD and DeD when faced with limited training data. While a reduction in test performance is observed, DistDeD is more robust to the removal of training data, in comparison to DeD. The maximum AUC value for each setting of reduced training data is provided in Table 8.

| | Percentage of Training Data | | | | |
| --- | --- | --- | --- | --- | --- |
| | 10% | 25% | 50% | 75% | 100% |
| DistDeD | 0.6848 | 0.7056 | 0.6988 | 0.7022 | 0.7912 |
| DeD | 0.5350 | 0.6075 | 0.6444 | 0.6438 | 0.6629 |

Table 8: Maximum AUC values evaluating DistDeD vs DeD performance when training in low-data regimes.

