# OpenReview forum: "Risk Sensitive Dead-end Identification in Safety-Critical Offline Reinforcement Learning"
_TMLR — Accepted by TMLR_

### Review · Reviewer_Cf5v · 2022-12-05

**Summary Of Contributions:**

The authors propose a framework to identify worst-case decision points by explicitly estimating distributions of the expected return of a decision.

**Audience:**

No

**Broader Impact Concerns:**

.

**Claims And Evidence:**

Yes

**Requested Changes:**

All my concerns are stated above.

**Strengths And Weaknesses:**

Strengths:
1. The experiment proves that  DistDeD significantly improves over prior discovery approaches, providing indications of the risk 10 hours earlier on average as well as increasing detection by 20%.
2. The author proposes a general-purpose framework that can be applied to a number of safety-critical applications using risk-sensitive RL to provide an early warning of risk over possible future outcomes.

Weakness:
1. What do you mean by saying that "However, safe RL assumes a priori knowledge of what unsafe regions are, which is not always feasible in real-world safety-critical scenarios."?
2. The entire paper is not fully quoted. Many of the latest papers or classic papers are not quoted. For example, in the Safe and Risk-sensitive RL section of Related work, the authors seem to have the most recent literature on SafeRL from 2017, which is inadequate, and I suggest that the authors be able to research the most recent papers.
    For SafeRL:

             Reward Constrained Policy Optimization ICLR2019.
             CRPO: A New Approach for Safe Reinforcement Learning with Convergence Guarantee ICML2021.
             Constrained Update Projection Approach to Safe Policy Optimization. NIPS 2022.
    For Offline Safe:

             Constrained Offline Policy Optimization. ICML 2022.

    For Uncertainty:

             PESSIMISTIC BOOTSTRAPPING FOR UNCERTAINTY-DRIVEN OFFLINE REINFORCEMENT LEARNING. ICLR 2022.

    Others:

             Is Pessimism Provably Efficient for Offline RL? ICML2021.

3. The experimental setup of the paper is too simple, which very much calls into question the validity of this approach in a complicated domain.There also does not seem to be an adequate comparison with baseline.

4. The experimental setup of this paper is not sufficiently described by the authors, and I would like the authors to provide a detailed list of parameters in the appendix.

---

> ### Author Response · Authors · 2022-12-15
> **Discussion about SafeRL**
>
> >What do you mean by saying that "However, safe RL assumes a priori knowledge of what unsafe regions are, which is not always feasible in real-world safety-critical scenarios."?
>
> (and)
>
> >Many of the latest papers or classic papers are not quoted.
>
> We thank the reviewer for their feedback and would like to clarify details relating to Safe RL. Safe RL is a field of RL which builds upon constrained Markov Decision Processes (CMDP). It is assumed across the literature we have studied (including the references the reviewer provided) that the constraints or penalties used for enforcing safety are pre-determined or known prior to learning in the environment. These constraints are usually hand designed to specifically guide agent behavior to avoid unsafe regions of the state space. When these constraints are available in the environment and known ahead of time, constrained Markov Decision Processes and the majority of approaches developed in safe RL, including the ones mentioned by the reviewer, are applicable. For instance, CRPO (Xu et al, 2020) focuses on maximizing expected total rewards, while avoiding violating certain constraints imposed directly on the cost. The authors explicitly enforce these constraints through designing an objective function that reflects these. Similarly, Constrained Offline Policy Optimization tries to optimize a policy based on the CMDP framework where constraints are assumed to be defined apriori by a domain expert (Polosky et al, 2022) or otherwise. However, in reality, we often don’t know what these constraints should be or how to explicitly define regions of safety. Our method focuses on identifying these regions and flagging them as potentially unsafe to alert a user to avoid these situations. Specifically, we provide a more realistic and reliable dead-end discovery approach that accounts for environment stochasticity by implementing distributional RL for learning the value function. We have added these citations for safe RL in Section 2 as well as included a more detailed explanation, following this response, of how our work differs from safeRL. The other citations provided have been integrated into Section 3.

---

> ### Author Response · Authors · 2022-12-15
> **Regarding apparent simplicity of our experiments (1/3)**
>
> > The experimental setup of the paper is too simple… There also does not seem to be an adequate comparison with baseline.
>
> We are inclined to thank the reviewer for saying that our experimental setup comes across as simple. Our primary goal in writing was to clearly convey the results and their justification of the claims we made about our proposed DistDeD framework, following TMLRs emphasis on this point. We do recognize however that this attempt at narrative clarity perhaps buried the detailed experimental setup that communicates the sophistication of what we did with each analysis used to evaluate DistDeD.
> We’d like to walk through the experiments used in this paper and outline some more practical details about how they were performed. As outlined in the general response to all reviewers, this paper was constructed to demonstrate improvement over the previously published dead-end discovery method (DeD; Fatemi, et al 2021). We evaluate both DeD and DistDeD in the same environments as this previous paper to directly show the improvements made through the contributions of DistDeD. As DeD is the first and only dead-end discovery method,to our knowledge, that does not rely on predetermined constraints (e.g. with a CMDP definition) to establish safety, we felt it was the only appropriate and fair baseline comparison aside from ablations to DistDeD, as shown in Section 6.2.3.
>
> ## **LifeGate**:
> 	- 1 million transitions collected with random policy from random initial states
> 	- Figure 3: Trained D- and R- Networks with DeD and DistDed to convergence, using the randomly collected trajectories. Evaluated several state locations to visualize outputs of the value functions. Chose two representative states and selected only the D-Network for visual simplicity
> 	- Figure 4 (A and B): Evaluated hand-designed policies (two purposefully suboptimal ones to demonstrate how early DistDeD would signal the risk of reaching the learned dead-end region in comparison to DeD. Figure 4(C) quantifies this advantage by collecting ten-thousand additional trajectories, by executing the two suboptimal policies. The metric measured is the time differential between when a flag is raised by DeD or DistDeD and when the dead-end region is reached.

---

> > ### Author Response · Authors · 2022-12-15
> > **Regarding apparent simplicity of our experiments (2/3)**
> >
> > ## **Sepsis Treatment**
> > - We extract data from the most recent release of the MIMIC dataset (v.4.0) – comprising over 40,000 patient stays – ensuring to remove known confounding ICU treatment departments from our collection. With extensive exploratory data analysis and filtering to ensure we had gathered a patient cohort with sufficient treatment and observation information, we ended with 6,188 patients who developed sepsis while in the ICU. As described at the outset of Section 6, we center the trajectories used for this study to contain 72 hours of observation. This dataset is split into a training/validation/evaluation set with the distribution 75/5/20, ensuring that the same proportion of surviving to non-surviving patient trajectories (~13.5% in this cohort).
> >
> > - As data from electronic health records is irregular and sparse, we follow previous RL for Healthcare literature to learn a fixed-dimensional latent encoding that aggregates information over time (Killian, et al 2020). We choose to perform this encoding by learning a Neural CDE (Morrill, et al 2021), trained to accurately reconstruct the trajectory at each hourly time-step. We fine tune the NCDE using Bayesian Optimization over the validation set of the data over 100 trials, using the final best performing model to then encode all patient data in each of the training/validation/evaluation datasets. This process is outlined in Section 6 as well as Section A.2.1 of the Appendix.
> >
> > - With the NCDE encoded data, we then train the value functions (IQN for DistDeD), tuning the hyperparameters of which with Bayesian Optimization using the validation set using 100 trials. For comparisons with DeD, we used the parameterization reported by Fatemi, et al (2021). This procedure is outlined in Section 6 with more detail given in Section A.2.2 in the Appendix.
> >
> > - With trained value functions for DeD and DistDeD, we then use the evaluation dataset to demonstrate the improvements provided by DistDeD. Namely, we sought to quantify how much earlier DistDeD would signal patient deterioration, what effect a choice of VaR threshold ($\alpha$) had on performance, as well as measure the contribution of the different components we used to develop DistDeD (the introduction of a) distributional RL and use of the b) CQL penalty). To run these analyses:
> >
> >   - We gather the full return distributions of both the D- and R-Networks for each time step of the patient trajectories (min 12 steps, max. 72 steps; median. 42 steps) and each possible action (25) by sampling 1000 particles from the IQN networks.
> >
> >   - In all analyses of the full-return distributions throughout Section 6, we perform separate analyses for 50 different settings of the value-at-risk threshold $\alpha$ for computing the CVaR, linearly spaced from 0 to 1 (in most figures, we plot only a subset of these analyses for visual clarity).
> >
> >   - We evaluated DeD using their published thresholds for $\delta_D$ and $\delta_R$, for the analysis in Section 6.2.1, we determined a single set of thresholds for DistDeD following the same analysis done by Fatemi, et al (2021). We highlight how this is done in Section A.4.1. In essence, we plotted sets of histograms (one for surviving patients, one for nonsurviving patients) of the computed CVaR for both the D- and R-Networks over all states for each time step, for all $\alpha$ values. We then attempted to select $\delta_D$ and $\delta_R$ that would separate the nonsurviving patient values from the surviving patient values, minimizing as many false positives (values from surviving patients that fall below the thresholds). Using these thresholds (for both DeD and DistDeD), we determine the first time step when the median of the CVaR values over all actions fell below the thresholds, for both D- and R-Networks respectively, which signifies a significant risk of the patient reaching a dead-end. In Figure 5, we measure how far ahead of patient death, in the case of non-surviving patients, this first flag is raised. The box plots are taken over the setting of CVaR risk threshold $\alpha$. Figure 6 represents the spirit of our empirical analyses and and we compare the temporal difference between DistDeD and DeD directly for all patients. We see again that DistDeD provides earlier indication of patient health deterioration, particularly as lower values of $\alpha$ are selected. This analysis introduced an important question about the balance between early warning and increased false positives.

---

> > > ### Author Response · Authors · 2022-12-15
> > > **Regarding the apparent simplicity of our experiments (3/3)**
> > >
> > > ## **Sepsis Treatment** (cont.)
> > > To address the concern of increased false positives, as well as to evaluate the range of possible settings for $\delta_D$ and $\delta_R$, we set out to define the notions of true-positive and false-positive within dead-end discovery (see Section 6.2.2). This also enabled us to establish a receiver operating characteristic (ROC) as a metric to holistically evaluate the performance of DistDeD vs. DeD. We evaluated each patient trajectory and aggregated the rate of nonsurviving patients having been correctly flagged by either DeD or DistDeD as well as the rate of surviving patients “wrongly” flagged. We construct the ROC curve by evaluating the sensitivity of the dead-end discovery process in each DistDeD and DeD by varying the thresholds $\delta_D$ and $\delta_R$ to provide a more complete picture of the performance of any method. Figure 7 shows this comparison, allowing us to see that DistDeD robustly outperforms DeD, regardless of the choice of $\alpha$ (each green curve corresponds to an independent setting).
> > >
> > >
> > > In Section 6.2.3, we repeat all of the above training and evaluation paradigms for two ablations of DistDeD, by removing either the distributional component (essentially running DeD with a CQL penalty, which could be thought of as a separate baseline) or the CQL penalty. We present in Figure 8 the summary of this evaluation using the quantitative measure of Area under the ROC curve as a comparison. Table 1 takes the maximum AUC of each approach (picking the best configuration of $\alpha$ for DistDeD and DistDeD without CQL). Here, we conclude that DistDeD provides a 20% improvement over DeD using this AUC metric.
> > >
> > >
> > > In response to Reviewer `NY6s` we also ran additional experiments where we:
> > >
> > >  1) evaluated increasing levels of conservatism (by increasing the weight of the CQL penalty during optimization). This required training additional models following the established procedures and analyses above. See Section A.4.3.
> > >
> > >  2) investigated the stability of the distributional RL in low data regimes. We recommend our response to Reviewer NY6s for more context. But, in this experiment we subsampled the training set, while maintaining the proportions of surviving/nonsurviving patients, to the following percentages of the original training set 10%, 25%, 50%, 75%. Using these reduced datasets, we retrained DistDeD and DeD and then evaluated using the held out test set, using the maximum AUC across all $\alpha$ settings for DistDeD, we see that DistDeD maintains performance superiority in these low data regimes. See Section A.4.4
> > >
> > > **We hope that by listing out these practical details of our experiments we have been able to demonstrate to the reviewer of the thorough nature by which we have sought to validate DistDeD in comparison to DeD.** Since we chose to focus our writing in this paper on the takeaways of improving over DeD, we purposefully did not want to distract from this by exhaustively making claims about the complexity or challenge of the validation using the toy example, EHR data processing, the NCDE encoding as well as the minutiae of training + evaluating the value networks. Thanks to the comments we received, we realize that several of these points are necessary to demonstrate the sophistication of our methodology. We have added additional descriptors in Sections 5 and 6 that will help clarify this as well as adding a new section (A.3) to the Appendix to help elucidate these details.

---

> ### Author Response · Authors · 2022-12-15
> **"The experimental setup of this paper is not sufficiently described by the authors"**
>
> We attempted to provide these details in both Sections 5 (first paragraph under “Empirical comparison”) and 6 (references to Appendix Section A.2) but realize, thanks to your comment we see that our efforts were not sufficient. We have added a new Section to the appendix (Section A.3) that outlines the specific details of each experiment and model setting which also summarizes the listing provided in the previous comment. **Thank you for this recommendation.**

---

> ### Author Response · Authors · 2023-01-13
> **Seeking discussion**
>
> We're hopeful that we've addressed all of the concerns raised by the reviewer. Primarily, we are interested in discussing whether our responses have satisfactorily improved the reviewer's evaluation of our work and if there is any additional clarification needed. This was the only review that had a negative response to the "Audience" question and we hope that the reviewer's opinion has changed positively.
>
> To quote from the general comment we posted on 15 December:
>
> > We believe that this paper is of high relevance and interest to the machine learning research community, in particular those considering solutions to real-world problems. From an application perspective, in a safety-critical setting such as medicine, our approach serves as an early warning system that can alert a domain expert up to 28 hours in advance of adverse events such as patient death, enabling timely, potentially life-saving intervention. In comparison, newer real-time warning systems based on the classification of Sepsis onset, such as TREWS (Henry et al 2015, Adams et al 2022), can do so anywhere between 3 and 22 hours in advance. Beyond healthcare, we see our approach as important for building robust systems that can anticipate risk in advance, such that these risks can be mitigated. From the methods perspective, our approach integrates ideas from distributional RL with a framework for identifying dead-ends to learn a lower bound on the return. Further, we incorporate an additional conservative Q-learning penalty to enable avoiding overestimating the values of actions for which we don’t have enough evidence in our dataset. While each of these ideas have been developed separately and applied for different purposes, they have not been integrated into a unified framework as we propose here.

---

### Review · Reviewer_DnCM · 2022-12-05

**Summary Of Contributions:**

Below there is my initial message to the author, left for the reason that the subsequent answers are in context.

The paper addresses the problem of risk-sensitive decision making. The problem is important when safety issues are critical (e.g. estimating outcomes of sepsa treatment). The particular notion is 'dead-end identification', namely, understanding states/transition, which lead to unrecoverable areas.



<initial message>
Dear authors,

I find some parts of the paper confusing enough so that it is hard for me to write a meaningful review. This is, perhaps, due to my lack of understanding of the studied setting or being somewhat beyond my expertise. I would be happy to come back to the task once the questions below are clarified.

1. what is really the setting, is it an MDP $\mathcal{M}$ with $R\in$ {-1, 1}. This seems to be suggested in Sec 4, but never said explicitly
2. what is "immediate negative termination" (definition of $F_D$) and how is it different from transition to the dead-end (definition of $P_D$)
3. The last paragraph of Sec 3 defines "dead-end" is it the notion which is valid in the whole paper? Also in the distributional setting?
4. Take $\mathcal{M}_D$ (sec 4), from what is said about $\mathcal{R}_D$ and the fact that the support of $Z_D$ is $[-1,0]$ I assume that the MDP terminated after receiving $\mathcal{R}_D = -1$. Is it correct?
	- if yes, then in fact the support of $Z_D$ is {-1,1} (the paper assumes $\gamma=1$); the distribution is discrete. In this case, the assumed distributional framework is not relevant, as it models continuous distributions.

**Audience:**

Yes

**Claims And Evidence:**

Yes

**Requested Changes:**

Clarify the above questions.

**Strengths And Weaknesses:**

Strengths:
 - important and valid problem of practical relevance
 - the work is on the particularly hard intersection of off-line RL and safety
 - real-life data analysis

Weaknesses:
 - the novelty of the paper is somewhat limited, the binary outcomes might be limiting in some applications
 - the provided experimental results, although interesting, make it hard to assess fully if the method is robust
 - I suspect, that due to the specific setting (binary outcomes) some easier approaches might be applicable. For example, a direct estimation of the probability of entering the dead-end region.

---

> ### Author Response · Authors · 2022-12-15
> **Answering questions**
>
> **We are disappointed that the reviewer was not able to fully evaluate our paper. We are grateful for the questions raised and hope that we can clearly address them here, in a way that enables an adequate review of our work.**
>
> >What is really the setting, is it an MDP with R… This seems to be suggested but never said explicitly
>
> Yes, following (Fatemi, et al; 2021) we consider MDPs with sparse binary reward structure, where positive outcomes are rewarded with +1 and negative outcomes receive -1. We have made this explicit at the beginning of Section 3 and reaffirmed this at the beginning of Section 4.
>
> >What is immediate negative termination ($F_D$) and how is it different from transition to dead-end ($P_D$)?
>
> $F_D(s,a)$ corresponds to the probability that taking action a from state s transitions directly to the negative terminal state, this is what is meant by “immediate negative termination”. This differs from the transition to a dead-end state which, by definition, is any intermediate state from which a negative terminal state is guaranteed to occur. This negative termination may occur after some random number of transitions. We have clarified this in the “Dead-end Discovery” paragraph of Section 3. The security condition introduced by (Fatemi, et al; 2021) considers both possibilities of applying action $a$ in state $s$ as things to avoid and designs the dead-end discovery (DeD) framework to satisfy this condition. We build directly from this foundation as described in Section 4, providing a bound on this satisfying quantity using the conditional value at risk.
>
> >The last paragraph of Section 3 defines a dead-end  is it the notion used in the whole paper? Also in the distributional setting?
>
> Yes, we used the same definition of dead-end throughout the whole paper as we improve directly from (Fatemi, et al; 2021). We adapt the designation of a state being a dead-end just slightly when using a distributional framework, but the overall definition of a dead-end is the same. Please see the paragraph in Section 4 that directly follows Figure 2 that provides the description of how our proposed distributional dead-end discovery (DistDeD) framework is used to designate dead-ends. We have clarified this in both Sections 3 and 4.
>
> >Take M_D from what is said about R_D… I assume that MDP terminates after receiving R_D = -1. Correct? If yes, then the support of Z_D is {-1, 1}; the distribution is discrete and the assumed framework is not relevant.
>
> Yes, the assessment is correct of the MDP $\mathcal{M}_D$ terminating when a negative reward is received (indicating a negative terminal outcome is reached). But, we would like to remind the reviewer that all positive terminal conditions of the original MDP $\mathcal{M}$ are still present, but in $\mathcal{M}_D$ the reward for reaching them is zero. We’re happy that the reviewer understood that we do not use discounting (e.g. $\gamma =1$) but fear that there is a misunderstanding of the real-world safety-critical environments we have considered in the development of DistDeD, following the foundation laid by (Fatemi, et al; 2021). Specifically, the healthcare setting we consider is not deterministic in a way that would provide the discrete value distribution (which would be {-1, 0} for $\mathcal{M}_D$ and {0, 1} for $\mathcal{M}_R$). The transition and outcomes from similar patient states may vary stochastically–admitting a possible range of values between -1 and 0–justifying the use of the continuous distributional framework we have used as a basis for DistDeD. At a high-level, the dead-end discovery framework (introduced by Fatemi, et al and improved upon in this work) indicates whether there is an increased likelihood of reaching a dead-end (and thereby negative terminal state) when considering each action. If that likelihood surpasses an empirically determined threshold, that action is recommended to be avoided. We have clarified the types of environments we assume the dead-end discovery framework is suitable for at the beginning of Section 3.
>
>
>
> _Fatemi, M., Killian, T.W., Subramanian, J. and Ghassemi, M., 2021. Medical Dead-ends and Learning to Identify High-risk States and Treatments. Advances in Neural Information Processing Systems, 34, pp.4856-4870._

---

> ### Author Response · Authors · 2023-01-11
> **Response to updated review**
>
> We thank the reviewer for taking the time to re-read and evaluate our submitted paper. We are grateful that the reviewer recognized the difficulty and importance of the problem domain we have approached in this work that has sought to improve on a promising paradigm (Dead-end Discovery; Fatemi, et al 2021) for using offline RL with real-world datasets. For concerns about novelty, we refer to our General Comment made to all reviewers on 15 December 2022.
>
> In terms of the limitations mentioned here. We acknowledge the limitations of requiring sparse binary outcomes in Section 7. Without this restriction, the theory underlying dead-end discovery fails to provide guarantees that the value functions adequately communicate the risk of a state entering an irrecoverable dead-end. The expansion of this theory to encompass dense and more varied reward functions is a point of future work. We will make this more clear in our final updated draft of the paper ahead of publication.
>
> We do agree that the use of the word "robust" in the title is misleading. Thank you for pointing this out. The origin of our thinking around "robustness" was in the level of improvement over DeD from Fatemi, et al. We were concerned at the outset of this project that there may be settings where DeD would outperform the proposed DistDeD. This however was not the case as our experiments show, and motivated by Figure 7, we determined that DistDeD was a robust improvement over DeD. But, as the reviewer has pointed out this is not a full demonstration of the robustness of the approach. __In response, we propose amending the title of the paper to remove "Robust".__
>
> To address this final point about "easier approaches" to estimate the dead-end region. This consideration largely follows our response to the questions raised by Reviewer Cf5v about the use of contemporary SafeRL approaches that are based on constrained Markov Decision Processes. If it were possible to define _a priori_ the dead-end region, then simpler approaches would be amenable to this estimation process. However, this paper focuses on difficult real-world domains where constructing this definition is intractable. This is why we have built from the established dead-end discovery framework, using offline RL to estimate the risk of any state being a dead-end based on long term reward. Additionally, while this was not featured in this paper (see the final paragraph of Section 3) the use of RL frameworks within dead-end discovery allow for direct estimation of the risk of individual actions leading to dead-ends. This provides more tangible support than a simple risk score, which is the ultimate goal of the development and improvement of the framework provided in this paper.

---

### Review · Reviewer_NY6s · 2022-12-06

**Summary Of Contributions:**

This work studies offline reinforcement learning in safety-critical settings. In such settings, one may want to avoid “dead-ends” or worst-case outcomes, and this work aims to develop an approach to “dead-end discovery”, where the goal is simply to train an agent on offline data to avoid dead-ends. Building off of existing work, they approach this from the perspective of distributional RL, seeking to model the entire distribution of the returns rather than just the mean. Through experiments on a toy example they show that taking into account the entire distribution yields an improvement over simply taking into account the mean of the distribution, and show that this also leads to improved performance on real-world sepsis data.

**Audience:**

Yes

**Broader Impact Concerns:**

None.

**Claims And Evidence:**

Yes

**Requested Changes:**

- A more thorough investigation of the tradeoff between performance and additional levels of conservatism would be helpful. As stated above, as we make the behavior more conservative, we would expect some tradeoff with a decrease in performance in other aspects, but this is not investigated at all. I believe it is important that an experiment exploring this aspect, and comparing to the performance of (Fatemi et al., 2021), is included.
- The approach proposed in this work estimates an entire distribution rather than just the mean, as is done in (Fatemi et al., 2021). One would expect this to lead to a degradation in performance in low data regimes, as it is more difficult to estimate an entire distribution than just the mean. This should be investigated more thoroughly with an experiment in the low-data regime, showing what the loss in performance is.
- More discussion on the CQL penalty would be helpful.

**Strengths And Weaknesses:**

Strengths:
- The problem seems relevant—safety in RL is important, and has seen a significant amount of attention in the community.
- The experimental results demonstrate that the proposed approach does lead to more conservative performance, better identifying dead-ends (though the improvement is not too significant).

Weaknesses:
- The proposed approach is only a slight modification of the approach from (Fatemi et al., 2021), the only modification being that they rely on distributional RL instead of standard RL to handle the dead-end discovery, leading to slightly more conservative performance in theory and practice. Given this, the novelty of this work is fairly minimal.
- The experimental evaluation is rather brief, only including results on a toy environment and a single dataset. Furthermore, while it is shown that the proposed algorithm is more conservative than the algorithm of (Fatemi et al., 2021), it is not shown how this conservatism could affect performance on a task (for instance, it may be more conservative, but this could lead to decrease in other performance, e.g. total reward acquired).

---

> ### Author Response · Authors · 2022-12-15
> **"(the improvement is not too significant)"**
>
> We strongly disagree with this sentiment. Early identification of high-risk patient states and corresponding suboptimal treatment options is highly significant in acute clinical situations. It has been shown, when treating sepsis, that interventions within tight time frames (even as little as 10 minutes) after suspected onset decreases patient mortality (Gauer, et al; 2020). By providing an additional 10-12 hours of early warning of patient deterioration beyond what (Fatemi, et al; 2021) secured with DeD is substantial. Our proposed DistDeD approach also increases the precision of such estimates, as evidenced by the several analyses presented in our paper assessing the balance between the true positive and false positive rates of identifying dead-end patient states.
>
> _Gauer, R., Forbes, D. and Boyer, N., 2020. Sepsis: diagnosis and management. American family physician, 101(7), pp.409-418._
>
> _Fatemi, M., Killian, T.W., Subramanian, J. and Ghassemi, M., 2021. Medical Dead-ends and Learning to Identify High-risk States and Treatments. Advances in Neural Information Processing Systems, 34, pp.4856-4870._

---

> ### Author Response · Authors · 2022-12-15
> **Additional experiments**
>
> >The experimental evaluation is rather brief…  it is not shown how this conservatism could affect performance on a task… this could lead to decrease in other performance (e.g. total reward acquired).'''
>
> There are two factors contributing to the level of conservatism employed by DistDeD: 1) the choice of value-at-risk threshold ($\alpha$) and 2) the weight given to the CQL penalty applied to the IQN learning objective ($\beta$). By choosing either a smaller value for $\alpha$ or larger value for $\beta$, the conservatism is increased. **We treat the choice of these parameters differently.** The choice of $\alpha$ can be made in a post-hoc fashion after estimating the value distribution for a given state and action. Thereby the level of conservatism of the dead-end discovery heuristic can be tuned according to the task. We demonstrate the effect of different $\alpha$ values throughout Section 6 in the paper. Since $\beta$ directly influences the objective function used to train the D- and R- networks, we treat it as a hyperparameter to be tuned during training. _We have clarified the use of $\alpha$ and $\beta$ as levels of conservatism in Section 4 and are grateful for the suggestion to do so by this comment._
>
> We neglected to demonstrate the effect larger values of $\beta$ has on the performance of dead-end discovery with DistDeD and are grateful for the recommendation to provide this analysis in our paper. We have included a new subsection in the appendix (Section A.4.3) that will be referenced from Section 4 which contains this analysis that clearly shows the seemingly negative effects an increased value for $\beta$ has on the performance of DistDeD across all choices of $\alpha$. With an increase of $\beta$ to 0.2, we see that half of the values of $\alpha$ have performance worse than (Fatemi, et al; 2021). While other metrics of performance, such as learning a policy, are out of scope for this work (see the General comment made to all reviewers) we agree that high levels of conservatism would negatively impact top-line performance as a trade-off as those other metrics are within scope of the data + desired task.
>
> >One would expect this to lead to a degradation in [DistDeD] performance in low data regimes…This should be investigated more thoroughly with an experiment in the low-data regime
>
> As discussed in the General comment made to all reviewers, the motivation stated by (Fatemi, et al; 2021) for the development of the dead-end discovery framework was in response to safety-critical settings in low data regimes and non-exploratory learning paradigms. It is a natural question to wonder if a seemingly more complex object such as an implicit distribution would struggle in such settings. However, prior works leveraging distributional RL in low data regimes have shown that the distributional counterpart does not experience a collapse in performance in comparison to non-distributional algorithms (see Figure 6 in (Agrawal, et al; 2020) and Table 3 in (Kumar, et al; 2020), each building from the QR-DQN (Dabney, et al; 2018)). Yet, to address this question in our safety-critical offline setting we re-trained both DistDeD and DeD (Fatemi, et al; 2021) with a random subset of the training data, ensuring that the proportion of negative and positive trajectories is maintained. We trained separate value functions using 10%, 25%, 50% and 75% of the training data. We evaluated each experiment using the same test dataset and procedures seen in Section 6.2. This analysis is contained in a new subsection in the Appendix (Section A.4.4, specifically Figure 11 and Table 2) _which is referenced in Section 6_. We confirm the demonstrations of prior literature, where distributional RL avoids an outsized reduction in performance to its non-distributional counterparts. **Specifically, DistDeD maintains performance superiority over DeD as the low-data regimes become more severe.**
>
> _Fatemi, M., Killian, T.W., Subramanian, J. and Ghassemi, M., 2021. Medical Dead-ends and Learning to Identify High-risk States and Treatments. Advances in Neural Information Processing Systems, 34, pp.4856-4870._
>
> _Agarwal, R., Schuurmans, D. and Norouzi, M., 2020, November. An optimistic perspective on offline reinforcement learning. In International Conference on Machine Learning (pp. 104-114). PMLR._
>
> _Kumar, A., Zhou, A., Tucker, G. and Levine, S., 2020. Conservative q-learning for offline reinforcement learning. Advances in Neural Information Processing Systems, 33, pp.1179-1191._
>
> _Dabney, W., Rowland, M., Bellemare, M. and Munos, R., 2018, April. Distributional reinforcement learning with quantile regression. In Proceedings of the AAAI Conference on Artificial Intelligence (Vol. 32, No. 1)._

---

> ### Author Response · Authors · 2022-12-15
> **"More discussion on the CQL penalty would be helpful"**
>
> Thank you for this recommendation. We agree that a more complete description of CQL in Section 3 will help to improve the paper and clarify its use in DistDeD to reduce overestimated values of actions not presently seen in the dataset and thereby increasing the conservatism of the trained value functions. _We have expanded the “Conservatism in Offline RL” paragraph to include this._ This will help prepare the reader to follow the discussion proposed above about the trade-off between increased conservatism and performance.

---

> ### Comment · Reviewer_NY6s · 2023-01-11
> **Response to authors**
>
> I would like to thank the authors for their comments and the additional experiments. I believe the majority of my concerns were addressed and do not have additional questions at this time.

---

> > ### Author Response · Authors · 2023-01-11
> > **Thank you**
> >
> > We'd like to thank the reviewer for their suggestions they provided. They certainly helped to improve the paper and clarify its contributions

---

### Author Response · Authors · 2022-12-15
**General response to initial reviews**

As a general comment to all reviewers we want to walk through the framing of the dead-end discovery framework introduced by Fatemi, et al (2021) and discuss our intended framing by submitting our work for consideration for publication within TMLR. We found the dead-end discovery (DeD) framework to be intriguing as it introduced a novel way of considering the use of Reinforcement Learning in offline safety-critical *and* data-limited settings, where learning and evaluation of an optimal policy is intractable. As we studied DeD, we recognized critical flaws in the notion of risk (an empirically determined threshold over a point estimate) used to designate which actions should be avoided and the eventual labeling of states as dead-ends. With our proposed distributional dead-end discovery (DistDeD), we have sought to improve on DeD by implementing a conservative, risk-sensitive evaluation approach made possible through  the estimation of the full distribution of return. We do not attempt or claim anywhere in our work to have developed a _novel_ algorithmic framework. **This however does not preclude the significance of the contributions we have provided with the introduction of DistDeD.** As stated in the paper, we have enabled earlier detection of dead-end states with a flexible framework that can be tuned according to the preferences of the human decision maker interacting with a hypothetical deployment of DistDeD. _Given the stated mission of TMLR to “emphasize technical correctness over subjective significance [or novelty]”, we felt that this was the appropriate venue to publish our improvements for dead-end discovery._

The original framing of dead-end discovery, as a means by which to identify actions and states to avoid, precludes any proactive decision making and as such, is not to be confused with learning a policy. RL is made up of two parts, prediction and control (Sutton and Barto; 2018), that are often pursued in tandem but are unique problems on their own. Prediction is focused on the estimation of the value function given some policy (exactly this line of work) where control is centered on approximating optimal policies from a value function. When learning a value function in a fully offline and off-policy setting, we build from the foundations of DeD to solely focus on the prediction problem since learning an optimal policy may be fraught due to data limitations and inadequate off-policy evaluation techniques. We refrain from making claims that we learn an optimal policy in order to avoid assuming algorithmic performance that cannot be tested in any satisfactory manner. We contend that RL can still be useful in these settings through the use of value functions.

Overall we believe our paper is of high relevance and interest to the machine learning research community, in particular those considering solutions to real-world problems. From an application perspective, in a safety-critical setting such as medicine, our approach serves as an early warning system that can alert a domain expert up to 28 hours in advance of adverse events such as patient death, enabling timely, potentially life-saving intervention. In comparison, newer real-time warning systems based on the classification of Sepsis onset, such as TREWS (Henry et al 2015, Adams et al 2022), can do so anywhere between 3 and 22 hours in advance. Beyond healthcare, we see our approach as important for building robust systems that can anticipate risk in advance, such that these risks can be mitigated. From the methods perspective, our approach integrates ideas from distributional RL with a framework for identifying dead-ends to learn a lower bound on the return. Further, we incorporate an additional conservative Q-learning penalty to enable avoiding overestimating the values of actions for which we don’t have enough evidence in our dataset. While each of these ideas have been developed separately and applied for different purposes, they have not been integrated into a unified framework as we propose here.

We address each specific question or point of concern raised by each reviewer in separate comments below. We are grateful for the provided recommendations to improve the clarity and overall quality of the paper. We have revised our paper as recommended by the reviewers to clarify points of confusion or otherwise improve the connection between preliminary concepts and the development of our proposed improvement for dead-end discovery in safety-critical offline RL settings. We have also included two new experimental analyses, as suggested by Reviewer `NY6s`, in the Appendix. All updates to the paper are printed in blue.

---

> ### Author Response · Authors · 2022-12-15
> **Corresponding citations**
>
> _Fatemi, M., Killian, T.W., Subramanian, J. and Ghassemi, M., 2021. Medical Dead-ends and Learning to Identify High-risk States and Treatments. Advances in Neural Information Processing Systems, 34, pp.4856-4870._
>
> _Sutton, R.S. and Barto, A.G., 2018. Reinforcement learning: An introduction. MIT press._
>
> _Henry, K.E., Hager, D.N., Pronovost, P.J. and Saria, S., 2015. A targeted real-time early warning score (TREWScore) for septic shock. Science translational medicine, 7(299), pp.299ra122-299ra122._
>
> _Adams, R., Henry, K.E., Sridharan, A., Soleimani, H., Zhan, A., Rawat, N., Johnson, L., Hager, D.N., Cosgrove, S.E., Markowski, A. and Klein, E.Y., 2022. Prospective, multi-site study of patient outcomes after implementation of the TREWS machine learning-based early warning system for sepsis. Nature medicine, 28(7), pp.1455-1460._

---

### Author Response · Authors · 2022-12-23
**Eagerly waiting further interactions**

After a week of submitting our responses to the initial reviews and preparing a revised version of the submitted paper, we'd like to signal our interest in having continued discussions with the reviewers. We would appreciate the opportunity to further clarify any aspect of our work as well as answer additional questions the reviewers may have.

Primarily, we are hoping to hear whether our responses and the revised paper have satisfied the concerns + requests from the reviewers.

Thanks!

---

> ### Comment · Action_Editors · 2023-01-03
> **Delays likely**
>
> Hello, because of the holiday season it is likely that the reviewers were not checking for rebuttals. I will solicit comments from the reviewers and give additional time for feedback before moving to the official decision.

---

> > ### Author Response · Authors · 2023-01-03
> > **That's understandable**
> >
> > Thank you!
> >
> > We've been grateful for the engagement so far and have recognized that we submitted at a tough time with several holidays and conferences. We are hopeful that our responses have been sufficient to prompt good discussion as well as help clarify points of concern that the reviewers raised.

---

### Decision · Action_Editors · 2023-01-22

**Recommendation:** Accept with minor revision

**Comment:**

The authors made particular commitments to reviewers in their response, in particular with regard to changing the title. I would like to see these changes committed before the final version is published.

**Audience:**

This paper takes a step on the critical path towards applying (offline) RL methods to real world tasks, and as such will be of interest to at least some of the TMLR readership.

**Claims And Evidence:**

This paper deal with learning from offline policy data in risk-sensitive domain so as to avoid producing agents that risk reaching irrecoverably negative condition, The reviewers found the contribution interesting an meaningful, and modulo some claims about the robustness of the method which need to be tempered (so as to show where future work could address key limitations), interesting enough to warrant publication. The sole dissenting voice amongst reviewers specified that this is not their field of expertise. Aside from (slightly subjective, as always) objections about novelty, their review doesn't really rule out the paper having impact, and the author response was sufficiently detailed and convincing that I am happy to lean towards recommending publication.

---

> ### Author Response · Authors · 2023-01-25
> **Thank you!**
>
> We are very grateful for this positive news! We have already taken the steps promised in our responses to improve the paper as evidenced by the most up-to-date PDF (including a change in title). We will however take a more thorough read toward the end of this week to finalize the camera ready, focusing on tempering claims of robustness and fleshing out the corresponding limitations.
>
> Thank you again!

---

> > ### Comment · Action_Editors · 2023-01-28
> > **Please update me when the camera ready is posted, with a summary of the changes made**
> >
> > What the title says ☝️

---

> > > ### Author Response · Authors · 2023-01-30
> > > **Summary of changes made for camera-ready**
> > >
> > > As agreed upon following feedback from the reviewers, we have made the following changes to prepare our camera ready paper:
> > >
> > >  - We have added the citations recommended by Reviewer `Cf5v` and included a more thorough discussion about safeRL and constrained MDPs in Section 2. We have clarified why these approaches are not suitable for the use cases explored in this paper.
> > > - Clarified the use of $\alpha$ and $\beta$ in terms of the level of conservatism they apply to the DistDeD framework in Section 4.
> > >  - We added more discussion on the CQL penalty in Section 3.
> > >  - We have clarified the MDP formulation that we build DistDeD around in Sections 3 and 4, explicitly mentioning the types of environments we expect dead-end discovery to be useful for, that we assume a sparse binary reward, and further clarify the security condition.
> > >  - We clarified the definition of “dead-end” used in this paper in Sections 3 and 4, connecting directly to the definition used in Fatemi, et al (2021)
> > >  - We added further details behind the experiments used in this paper in Sections 5 and 6. We also added a new section (A.3) to the Appendix devoted to walking through the setup of each experiment, providing detailed summaries of the parameters and hyperparameters used for each.
> > >  - We added additional experiments to 1) evaluate the effects of increased conservatism (focusing on the $\beta$ hyperparameter) and 2) evaluate whether the distributional framework was susceptible to catastrophic performance decay with reduced training data. Both of these experiments were added to new subsections in the Appendix (Sections A.4.3 and A.4.4 respectively)
> > >  - We have clarified the limitations of the requirement for dead-end discovery to only operate on sparse binary rewards.
> > >  - We have removed any claims or discussion points that would misconstrue the overall robustness of DistDeD. This includes updating the title of the paper to remove the word “robust”.
> > > - We added links to our public repository for the code used in this work
> > > - We added both a Contributions and Acknowledgments section at the end of the paper.
> > >
> > > Thank you again for helping us in this process. We are thrilled to have this paper published!

---

> > > > ### Comment · Action_Editors · 2023-01-30
> > > > **LGTM**
> > > >
> > > > Thanks for the breakdown of changes, and for reminding me I needed to look at this by posting a thread on Twitter 😅
> > > >
> > > > Looks good to go.